# ANIMAL BEHAVIORAL ANALYSIS AND NEURAL ENCODING WITH TRANSFORMER-BASED SELF-SUPERVISED PRETRAINING

**Yanchen Wang**[1*]**, Han Yu**[1*]**, Ari Blau**[1]**, Yizi Zhang**[1]**, International Brain Laboratory,**
**Liam Paninski**[1]**, Cole Hurwitz**[1] **& Matthew R. Whiteway**[1]
[1]Columbia University, New York, NY USA
[*]Equal contribution
`{yw4503,hy2562,bsb2144,yz4123,lmp2107,ch3676,mw3323}@columbia.edu`

## ABSTRACT

The brain can only be fully understood through the lens of the behavior it generates–a guiding principle in modern neuroscience research that nevertheless presents significant technical challenges. Many studies capture behavior with cameras, but video analysis approaches typically rely on specialized models requiring extensive labeled data. We address this limitation with BEAST (**BE**havioral **A**nalysis via **S**elf-supervised pretraining of **T**ransformers), a novel and scalable framework that pretrains experiment-specific vision transformers for diverse neuro-behavior analyses. BEAST combines masked autoencoding with temporal contrastive learning to effectively leverage unlabeled video data. Through comprehensive evaluation across multiple species, we demonstrate improved performance in three critical neuro-behavioral tasks: extracting behavioral features that correlate with neural activity, and pose estimation and action segmentation in both the single- and multi-animal settings. Our method establishes a powerful and versatile backbone model that accelerates behavioral analysis in scenarios where labeled data remains scarce.

## 1 INTRODUCTION

Understanding the relationship between brain and behavior is a fundamental challenge across a wide range of medical and scientific disciplines (Krakauer et al., 2017; Datta et al., 2019). Precise methods for extracting meaningful information from behavioral videos are essential for advancing these fields (Pereira et al., 2020). Self-supervised learning has revolutionized image and video understanding through large-scale foundation models (Chen et al., 2020; Caron et al., 2021; He et al., 2022), offering powerful tools that are beginning to transform scientific analyses (Huang et al., 2023a; Lastufka et al., 2024). However, these models have yet to be effectively translated to specialized domains like animal behavior analysis, creating a significant opportunity for methods that bridge cutting-edge machine learning with the specific demands of neuroscience and behavioral research.

Animal behavior videos present unique characteristics and challenges distinct from general video understanding. Controlled experiments generate large quantities of videos with static backgrounds and consistent camera angles, where the primary variation arises from animal movements and interactions. These videos enable numerous downstream analyses, and here we focus on three fundamentally different applications that collectively address a large proportion of behavioral neuroscience use cases: (1) neural activity prediction (or "neural encoding"), which requires extracting behavioral features that correlate with simultaneously recorded brain activity (Datta et al., 2019; Pereira et al., 2020; Urai et al., 2022); (2) pose estimation, which tracks specific anatomical landmarks for quantitative analysis of movement patterns (Mathis and Mathis, 2020; Pereira et al., 2020); and (3) action segmentation, which classifies distinct behavioral states like grooming, rearing or social interactions on every frame (Datta et al., 2019; Pereira et al., 2020). Each task demands different representations of the same underlying behavioral data, and current approaches typically require task-specific models and extensive labeled datasets (von Ziegler et al., 2021). Furthermore, most approaches fail to leverage the vast amounts of unlabeled data generated by behavior experiments, a significant untapped resource that, if harnessed properly, could substantially improve performance on these downstream tasks.

We address these challenges through a novel self-supervised pretraining framework for raw videos that produces a robust backbone for multiple downstream neuro-behavioral tasks. BEAST (**BE**havioral **A**nalysis via **S**elf-supervised pretraining of **T**ransformers) leverages the unique properties of experimental videos by combining masked autoencoding (He et al., 2022) to capture rich frame-level appearance information with temporal contrastive learning (Hyvarinen and Morioka, 2016) to model behavioral dynamics. We introduce a novel frame sampling strategy for the contrastive loss, designed to focus on learning representations of animal behavior against static backgrounds. BEAST trains on videos from a single experimental setup, creating tailored, versatile models that can be fine-tuned for multiple analytical needs specific to that experimental context. We demonstrate the value of this approach through comprehensive evaluation on three downstream tasks: (1) neural encoding in three mouse datasets; (2) pose estimation across four datasets spanning two species and single- and multi-view setups; and (3) action segmentation in both single- and multi-animal setups. BEAST achieves competitive or superior performance for the neural encoding and action segmentation tasks–outperforming baselines such as DINOv2 (Oquab et al., 2023)–while eliminating the need for the pose estimation step typically required by existing methods, dramatically reducing manual labeling effort. At the same time, pose estimation remains valuable for producing interpretable features critical to understanding movement dynamics, and BEAST enables significantly improved pose estimation for a given labeling budget. These results establish BEAST as a simple yet powerful foundation that can accelerate behavioral understanding across disciplines where fine-grained analysis is essential.

## 2 RELATED WORK

**Neural encoding models.** Neural encoding measures how observable signals predict neural activity, and provide a quantitative framework for interrogating neural representations. Earlier approaches applied generalized linear models to single neurons using controlled stimuli such as visual or auditory inputs (Paninski, 2004; Truccolo et al., 2005; Pillow et al., 2008; McFarland et al., 2013). More recently, deep learning methods have shown great promise in predicting neural population responses to sensory stimuli, including visual (Yamins et al., 2014; Schrimpf et al., 2018; Wang et al., 2025a), auditory (Kell et al., 2018; Li et al., 2023), and tactile (Zhuang et al., 2017) inputs. The widespread adoption of video monitoring during experiments has demonstrated that video-based behavioral covariates explain significant neural variability in both spontaneous (Stringer et al., 2019; Syeda et al., 2024) and task-driven behaviors (Musall et al., 2019; IBL et al., 2025a; Wang et al., 2025b; Chen et al., 2024; Zhang et al., 2025b). For example, Musall et al. (2019) showed that uninstructed movements explain a substantial fraction of cortical neural variance, and the International Brain Lab leveraged large-scale, region-resolved encoding analyses to chart the distribution of task-related information across the brain (IBL et al., 2025a). However, extracting rich spatiotemporal information from video remains challenging. Most studies rely on either a small set of keypoints (Syeda et al., 2024; IBL et al., 2025a; Wang et al., 2025b; Chen et al., 2024) or latent dimensions using PCA (Stringer et al., 2019; Musall et al., 2019) or autoencoders (Batty et al., 2019; Wang et al., 2025b; Chen et al., 2024), with limited efforts to predict neural activity directly from raw video (but see Wang et al. (2025b)).

**Large-scale models for behavioral video analysis.** Large-scale models for animal behavior analysis have predominantly focused on single tasks. For pose estimation, methods differ in how they balance flexibility and labeling requirements. DeepLabCut (Mathis et al., 2018) leverages ImageNet-pretrained backbones for fine-tuning on experiment-specific labeled datasets, offering flexibility but requiring more manual labeling. This work inspired a range of other general-purpose animal pose estimation tools including LEAP (Pereira et al., 2019), DeepPoseKit (Graving et al., 2019), TRex (Walter and Couzin, 2021), SLEAP (Pereira et al., 2022), and Lightning Pose (Biderman et al., 2024). In contrast, several specialized pose estimation tools provide tailored solutions for common experimental setups, such as top-down views of freely moving mice (Ye et al., 2024) and facial analysis of head-fixed rodents (Syeda et al., 2024; Daruwalla et al., 2024), significantly reducing labeling requirements. Similarly, in action segmentation, specialized systems developed for resident-intruder assays (Segalin et al., 2021; Goodwin et al., 2024) achieve high performance but remain limited to a specific experimental paradigm. While VideoPrism (Zhao et al., 2024) offers a general foundation model supporting multiple behavioral tasks (Sun et al., 2024), it relies on a frozen backbone trained on generic internet data rather than domain-specific content. Despite these advances, no accessible solutions exist for creating general behavior analysis models that leverage unlabeled data across multiple tasks. BEAST addresses this gap by enabling individual labs to develop experiment-specific models from their own unlabeled videos for diverse analyses.

**Self-supervised learning for images and videos.** Contrastive learning has emerged as a powerful self-supervised learning (SSL) framework; among its many predecessors and variants, SimCLR (Chen et al., 2020) popularized a simple and effective recipe that maximizes agreement between differently augmented views of the same sample via a contrastive loss in latent space. The contrastive method has also been extended to the temporal (Hyvarinen and Morioka, 2016) and video domains (Qian et al., 2021; Recasens et al., 2021; Dave et al., 2022). Another line of self-supervised approaches uses knowledge distillation, where a student network learns to match the outputs of a teacher network, such as DINO (Caron et al., 2021; Oquab et al., 2023; Siméoni et al., 2025). Masked modeling is a complementary approach that has demonstrated remarkable success, particularly masked autoencoding (MAE) (He et al., 2022), which revolutionized visual self-supervised learning by adapting BERT-style masked prediction to images using Vision Transformers (Dosovitskiy et al., 2020). VideoMAE (Tong et al., 2022) and BEVT (Wang et al., 2022) extended this approach to video data by leveraging spatiotemporal dependencies. Various works have combined contrastive and MAE objectives as a more efficient alternative for capturing spatiotemporal dependencies, as video models can require much more compute for training and inference (Mishra et al., 2022; Huang et al., 2023b; Lu et al., 2023; Lehner et al., 2024). Of note is VIC-MAE (Hernandez et al., 2024), which uses patch-based features for a masked autoencoding loss. The local features are also pooled into a global feature vector which is used with a contrastive loss computed across frames from multiple videos. The efficiency of contrastive-based methods compared to native video models is of particular interest in our application domain, where labs often do not have access to extensive compute resources. SSL techniques from computer vision have increasingly been adapted to the study of animal behavior. These approaches use SSL to extract useful feature representations from pose, image, or video data, which are then applied to downstream tasks such as action segmentation (Appendix F).

## 3 METHODS

BEAST uses a combination of an image-based masked autoencoding (MAE) loss–which excels at capturing per-frame appearance details–and temporal contrastive loss–which captures dependencies across frames (Fig. 1A). This integration enables a single backbone to excel across diverse downstream tasks, from precise keypoint localization to predicting complex structure in neural activity (Fig. 1B).

BEAST builds upon VIC-MAE (Hernandez et al., 2024), which combines masked autoencoding and contrastive losses, but introduces key adaptations for neuroscience applications. The most significant modification is how frames are sampled for the contrastive loss. VIC-MAE allows any two frames from the same video to be a positive pair, and frames from different videos are negative pairs (Xu and Wang, 2021). While this may be appropriate for benchmark datasets with short clips, animal behavior experiments generate long-duration recordings where behaviors repeat across time. We instead define positive frames within a narrow temporal window around the anchor ($\pm 1$ frame), while allowing negative frames to be either distant and dissimilar frames in the same video, or from different videos. Crucially, this strategy outperforms that of VIC-MAE (Table 5). See Appendix B for more details on our frame selection strategy and additional training and architecture simplifications of VIC-MAE.

**Vision transformer (VIT).** The standard image VIT (Dosovitskiy et al., 2020) data pipeline starts with a 2D image $\mathbf{x} \in \mathbb{R}^{H \times W \times C}$ ($H$, $W$, $C$ are height, width, channels) and splits it into 2D patches, each with shape $(P \times P \times C)$, where the patch size $P$ is typically 16. Each patch is reshaped to a vector of length $P^2 C$, and all patches are concatenated into a sequence of the $N$ flattened 2D patches $\mathbf{x}_p \in \mathbb{R}^{N \times (P^2 C)}$. Each flattened patch is mapped with a trainable linear projection to a "patch token," a vector of size $D$. We add 1D position embeddings to the patch tokens to retain patch location information. We add a learnable CLS token to the patch token sequence, which serves as a global representation for the image. The resulting patch tokens augmented with position embeddings ($\mathbf{t} \in \mathbb{R}^{N \times D}$) and the concatenated CLS token serve as the input to the standard VIT encoder.

**Masked autoencoding loss.** The masked autoencoding (MAE) loss randomly masks out a high proportion of the patch tokens (here, 0.75 (He et al., 2022)). We call the resulting unmasked tokens $\mathbf{t}_{um} \in \mathbb{R}^{L \times D}$, where $L = 0.25 \times N$ is the number of unmasked tokens. The unmasked tokens are processed by the VIT to produce embeddings $\mathbf{z}_{um} = \text{VIT}(\mathbf{t}_{um})$. The masked embeddings $\mathbf{z}_m \in \mathbf{R}^{(N-L) \times D}$ (consisting of all zeros) are then combined with the unmasked embeddings processed by VIT to form a complete patch sequence $\mathbf{z} \in \mathbb{R}^{N \times D}$, which is passed through a transformer decoder to produce a reconstruction $\hat{\mathbf{x}}_p \in \mathbb{R}^{N \times P^2 C}$ trained via mean square error: $\mathcal{L}_{\text{MSE}} = \frac{1}{N} \sum_{p=1}^{N} (\mathbf{x}_p - \hat{\mathbf{x}}_p)^2$. We refer to the model trained only with this MAE loss as VIT-M.

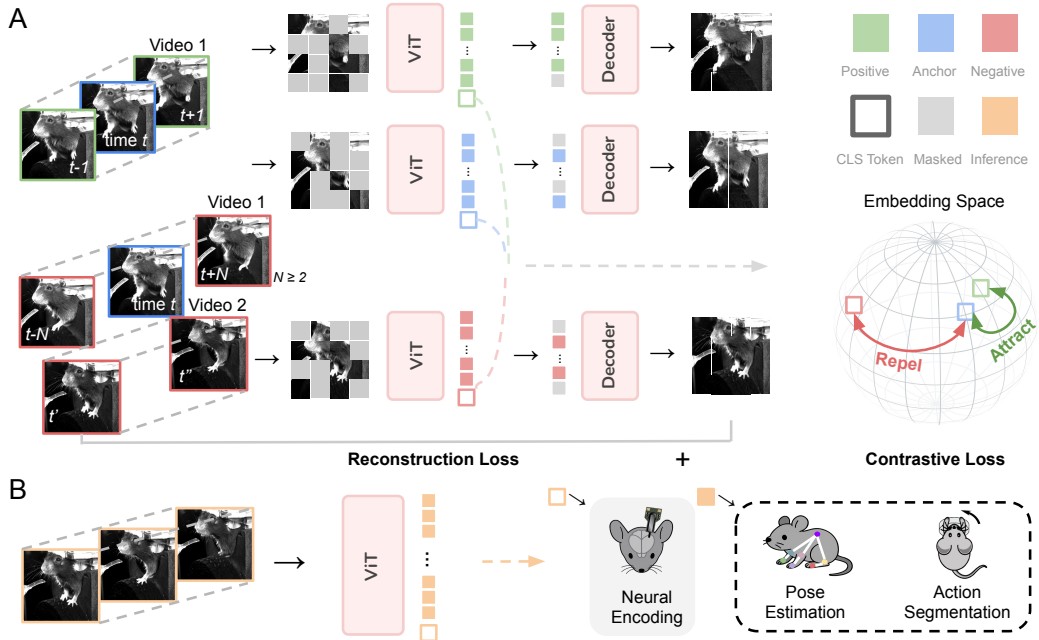

Figure 1: **BEAST framework. A:** Our self-supervised pretraining framework BEAST combines masked autoencoding (He et al., 2022) with temporal contrastive learning (Chen et al., 2020). An anchor frame at time $t$ is paired with a positive frame from $t \pm 1$, while more distant frames from the same video, or frames from other videos, serve as negative examples. Frames are divided into patches, with most patches randomly masked. A vision transformer (VIT) processes the remaining patches, which must reconstruct all patches. The VIT CLS tokens, which serve as a global representation of each frame, are nonlinearly projected to a new space where the contrastive loss pulls anchor-positive pairs together and pushes anchor-negative pairs apart. **B:** BEAST supports various downstream neuro-behavioral tasks including neural encoding, pose estimation, and action segmentation.

**Temporal contrastive loss.** The masked autoencoding loss is sufficient for reconstructing low-level features on individual frames. To imbue our embeddings with temporal information (which may be required for certain downstream tasks), we employ a contrastive loss that produces similar embeddings for frames close in time, and distinct embeddings for frames far apart in time or from different videos. To achieve this, each batch with $B$ samples contains $B/2$ anchor frames, and each anchor frame $\mathbf{x}_t^v$ (from time $t$ and video $v$) has a randomly chosen positive frame from $\mathbf{x}_{t\pm1}^v$. All remaining $B-2$ frames are treated as negative frames. Note this approach differs from other temporal contrastive losses that allow any frame from the anchor frame's video to be a positive frame (Xu and Wang, 2021; Hernandez et al., 2024), which does not perform well with temporally-extended behavioral videos (Table 5). To improve the robustness of this approach, we select the initial set of anchor frames from a given video to be as visually distinct from each other as possible (Table 4). We utilize the InfoNCE loss (Oord et al., 2018) computed on nonlinear projections of the CLS embeddings (which outperform other frame aggregation methods, Table 7) that are output by the VIT. The projector outputs $\{\mathbf{z}_b^p\}$ are used for the contrastive learning, calculated as $\mathcal{L}_{\text{InfoNCE}} = -\frac{2}{B} \sum_{i \in \mathcal{A}} \log \frac{\exp(\mathbf{z}_i^p \cdot \mathbf{z}_{i'}^p)}{\sum_{j \neq i} \exp(\mathbf{z}_i^p \cdot \mathbf{z}_j^p)}$, where $i'$ is the positive example associated with $i$ and $\mathcal{A}$ is the set of $B/2$ anchor frames. We refer to the model trained with both the masked autoencoding and contrastive losses as BEAST.

**Training and finetuning.** We initialize our models with pretrained ImageNet weights (Deng et al., 2009; He et al., 2022). Details of dataset construction, data augmentations, and batch construction are provided in Appendix B. We define the loss as $\mathcal{L}_{\text{MSE}} + \lambda \cdot \mathcal{L}_{\text{InfoNCE}}$, where $\lambda$ balances the two losses and is selected using the validation sets of the various datasets. Models are trained for 800 epochs using the AdamW optimizer (Loshchilov and Hutter, 2017) with a cosine annealing learning rate scheduler (Loshchilov and Hutter, 2016), taking approximately 25 hours on 8 Nvidia A40 GPUs.

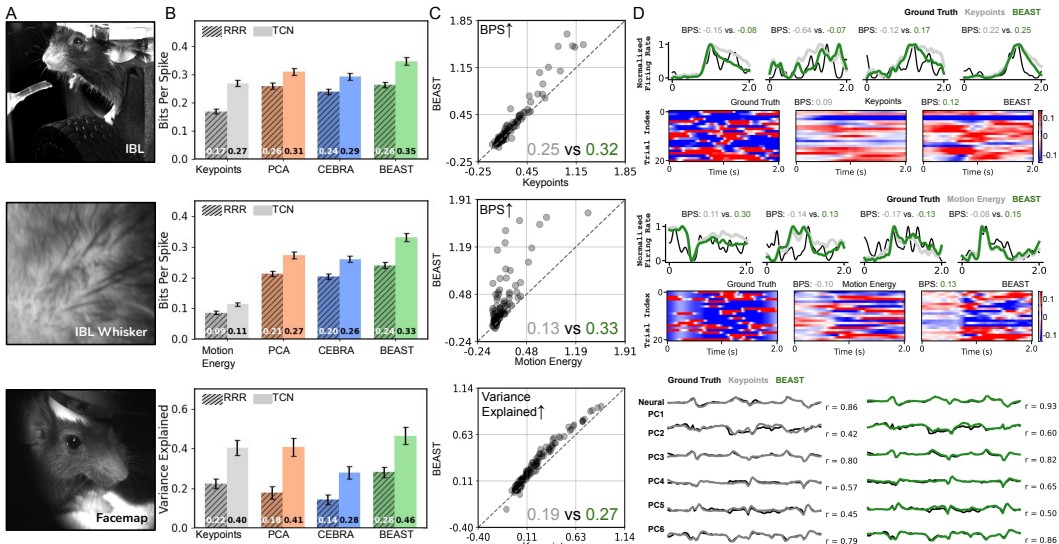

Figure 2: **BEAST improves neural encoding. A:** Example video frame from each dataset. **B:** Encoding performance is evaluated across multiple baseline features with both linear models (hatched bars; reduced rank regression, RRR) and nonlinear models (solid bars; temporal convolution network, TCN). CEBRA uses a contrastive loss to embed video frames in a latent feature space. "Motion energy" for the IBL-whisker dataset is a 1D estimate of movement calculated as the sum of the absolute pixel differences between successive frames. BEAST features outperform all baselines in both linear and nonlinear regimes. Error bars show standard error of the mean (S.E.M.) of Bits per Spike (BPS) across $N$=842 neurons from five test sessions (IBL and IBL-whisker) or S.E.M. of variance explained of the principal components of the neural activity across five test sessions (Facemap; see text). **C:** Scatterplot comparison of BEAST vs keypoint-based model performance in an example session. Each dot corresponds to an individual neuron. The values in the bottom-right corner represent the session-averaged BPS. **D:** *Top, middle*: comparison of the predicted trial-averaged firing rates for BEAST and keypoints (lines) and single-trial variability obtained by subtracting the neuron's average firing rate on each trial (heatmaps). *Bottom*: comparison of predicted neural principal components for the Facemap dataset.

## 4 RESULTS

We demonstrate the versatility of BEAST through comprehensive evaluation across three downstream neuro-behavioral tasks: (1) neural encoding, which challenges the model to extract spatiotemporal features that can predict patterns in neural activity; (2) pose estimation, which assesses the model's ability to extract fine-grained appearance details; and (3) action segmentation, which evaluates the model's capacity to extract spatiotemporal features required for predicting behavioral sequences. Throughout these evaluations, we present systematic ablation experiments that demonstrate the critical importance of our combined loss functions, and explore various adaptation strategies, including the use of CLS tokens or patch embeddings from a frozen backbone, as well as end-to-end fine-tuning.

### 4.1 NEURAL ENCODING

Predicting neural activity from behavior videos represents a significant challenge with promising implications for understanding the relationship between brain and behavior (Musall et al., 2019; Stringer et al., 2019; Wang et al., 2025b). Traditional approaches often rely on keypoints (IBL et al., 2025b; Syeda et al., 2024), potentially missing critical behavioral features that are not included in tracking or are obscured by fur or feathers. Studies have employed Principal Component Analysis on frames (Musall et al., 2019; Stringer et al., 2019), but this linear technique may inadequately capture subtle behavioral nuances. Transformer embeddings offer a compelling alternative, potentially outperforming linear approaches without being constrained by predefined keypoints, thereby providing richer representations that could reveal previously undetectable neuro-behavioral correlations.

**Datasets.** We present results on three high-quality neuro-behavioral datasets employing diverse neural recording technologies (Fig. 2). The first dataset is a head-fixed mouse performing a decision-making task the International Brain Laboratory (IBL et al., 2025a). This dataset, "IBL," features simultaneous behavioral video and neural activity monitoring at single-cell, single-spike resolution

using Neuropixels probes (Jun et al., 2017) spanning multiple brain regions (average of 168 neurons per session). The second dataset, "IBL-whisker," reuses the same sessions but utilizes a cropped area around the whisker pad in the video, a particularly salient behavioral feature for predicting neural activity (Stringer et al., 2019; Whiteway et al., 2021; Syeda et al., 2024). The third dataset comes from the Facemap study (Syeda et al., 2024), where neural activity is captured through two-photon calcium imaging, a technique capable of resolving a large number of individual cells, but unable to detect individual spikes. Following the authors' approach, we predict the principal components of the neural activity to capture predominant variance patterns across the large recorded neural populations.

**Models.** We first describe various feature representations used for neural encoding, followed by two models (linear and nonlinear) that we fit to each representation. The first representation for IBL and Facemap are keypoints tracked across the face and body (11 for IBL, 12 for Facemap). For the IBL-whisker dataset, which lacks keypoints, we instead utilize a 1D estimate of whisker pad motion energy (Appendix C). The second representation utilizes PCA applied to raw video frames, while the third leverages CEBRA (Schneider et al., 2023) (which employs a contrastive loss to embed inputs in a latent feature space) applied to raw video frames. Finally, we present results (using the CLS token) from BEAST, which is initialized with ImageNet weights then fine-tuned separately on each test session. Table 13 shows additional baselines that use frozen features from pretrained MAE (He et al., 2022), DINOv2 (Oquab et al., 2023), and CLIP (Radford et al., 2021) models. We train two encoders: linear encoders, which reveal how directly accessible information is within the features, and nonlinear encoders, which better determine the upper bounds of information content in the features. The linear encoder is a reduced rank regression model (Zhang et al., 2025a). The nonlinear encoder is the temporal convolution network (TCN) proposed in the Facemap study (Syeda et al., 2024).

**Evaluation.** All model hyperparameters are tuned to ensure robust baseline performance (Appendix C). To evaluate our neural encoding approaches, we utilize the Bits Per Spike (BPS) metric (Pei et al., 2021) on the spike-resolved IBL dataset (higher values better), and the $R^2$ metric on the neural principal components in the Facemap dataset. All models are evaluated on five test sessions.

**Results.** We find that nonlinear encoders consistently outperform their linear counterparts across all datasets and feature representations (Fig. 2 and Table 13). Notably, non-keypoint representations surpass keypoint-based approaches in both IBL and Facemap datasets, confirming our hypothesis that behavior videos contain richer information than what pose estimation typically captures. BEAST shows consistent improvements in neural encoding quality across all datasets and a range of dimensionalities (Fig. 8), indicating that BEAST's exceptional performance is not limited to high-dimensional embedding spaces. Interestingly, the comparable BPS values for BEAST in both IBL and IBL-whisker datasets suggest that a substantial portion of the neurally-relevant behavior information is captured by the whisker pad activity, at least in the recorded brain regions.

The VIT-based models in Fig. 2 are fine-tuned individually for each session to enable direct comparison with baseline approaches. We investigated whether pretraining provides additional benefits (Table 1). Strikingly, models pretrained on ImageNet using only the MAE loss, "VIT-M (IN)", outperform previous baselines without any fine-tuning. Further pretraining on 77

Table 1: Zero-shot neural encoding performance (BPS$\pm$1 standard error of mean over $N$=842 neurons across 5 test sessions).

| Method (TCN) | IBL | IBL-whisker |
|---|---|---|
| VIT-M (IN) | $0.321 \pm 0.013$ | $0.301 \pm 0.012$ |
| VIT-M (IN+PT) | $0.331 \pm 0.013$ | $0.311 \pm 0.013$ |
| VIT-C (IN+PT) | $0.314 \pm 0.013$ | $0.283 \pm 0.011$ |
| BEAST (IN+PT) | $0.292 \pm 0.012$ | $0.309 \pm 0.013$ |
| BEAST (IN+PT+FT) | $\mathbf{0.347 \pm 0.014}$ | $\mathbf{0.326 \pm 0.013}$ |

IBL sessions, "VIT-M (IN+PT)", improves performance on both IBL datasets, validating the importance of domain-specific pretraining. By incorporating the contrastive objective, BEAST achieves superior zero-shot performance: BEAST (IN+PT) outperforms the MAE-only variant, as well as a contrastive-only variant VIT-C. Session-specific fine-tuning, "BEAST (IN+PT+FT)", provides additional significant gains, reaching performance levels comparable to models fine-tuned directly from ImageNet weights (Table 13). Notably, even without fine-tuning, domain-specific pretrained models remain highly competitive, offering researchers a practical option when computational resources for fine-tuning are limited. Finally, we experimented with using the patch embeddings as input to the neural encoder, but found superior performance with the CLS tokens (Table 12).

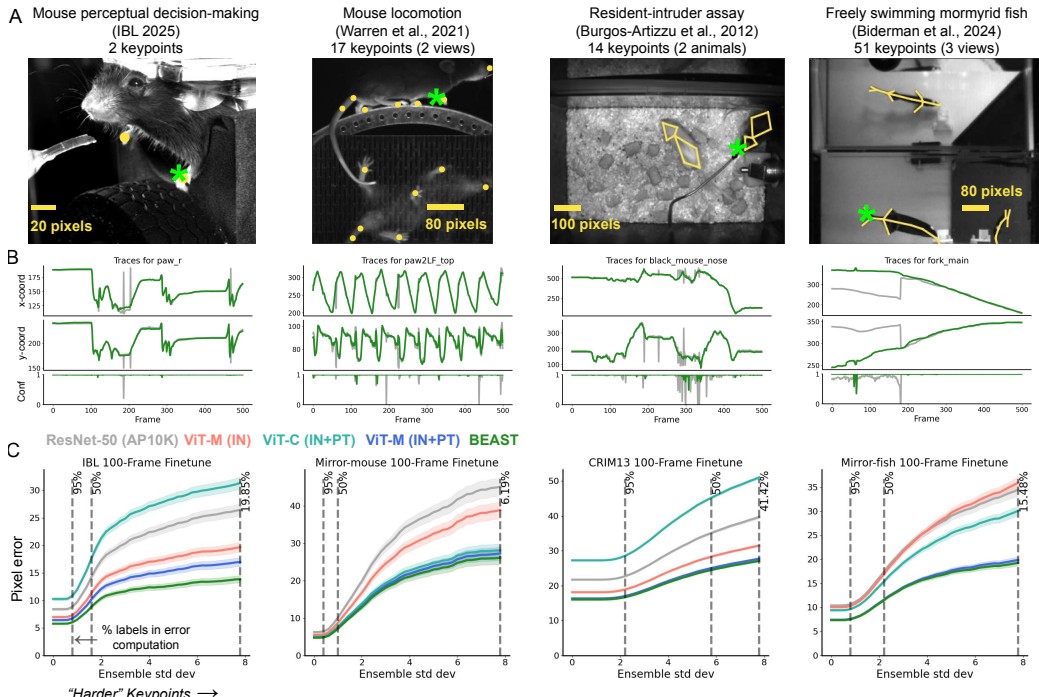

Figure 3: **BEAST improves pose estimation. A:** Example frame from each dataset overlaid with ground truth annotations. Green stars indicate the highlighted keypoint in panel B. **B:** Example traces from the ResNet-50 (gray) and BEAST (green) models for a single keypoint in a held-out video. BEAST traces evolve more smoothly in time and do not contain erroneous jumps like the ResNet-50 baseline. **C:** Pixel error as a function of keypoint difficulty (see main text; smaller is better): left-hand side shows performance across all keypoints; moving to the right drops the easier keypoints defined by inter-seed and -model prediction variance. Vertical dashed lines indicate the percentage of data used for the pixel error computation. VIT-M (IN) is a VIT backbone pretrained on ImageNet with a masked autoencoding loss; VIT-M (IN+PT) uses the same architecture and loss but is initialized with ImageNet-pretrained weights then further pretrained on experiment-specific unlabeled frames. VIT-C (IN+PT) performs the experiment-specific pretraining using the temporal contrastive loss only.

## 4.2 POSE ESTIMATION

Pose estimation is a fundamental technique in animal behavior analysis (Pereira et al., 2020), enabling precise quantification of posture and movement. Unlike human pose estimation, which benefits from extensive labeled datasets and standardized anatomy, animal pose estimation presents unique challenges such as scarcity of large annotated datasets and significant morphological diversity across species. Pretraining models on large volumes of unlabeled behavior videos can potentially reduce the labeled data requirements for accurate keypoint localization in various experimental paradigms.

**Datasets.** We present results on four distinct datasets (Fig. 3): (1) a head-fixed mouse performing a decision-making task (IBL et al., 2025a); (2) a head-fixed mouse running on a treadmill, seen from two views (Warren et al., 2021); (3) the Caltech Resident-Intruder Mouse (CRIM13) dataset, consisting of two freely interacting mice (Burgos-Artizzu et al., 2012); and (4) a freely moving weakly electric fish, seen from three views (Biderman et al., 2024; Pedraja et al., 2025).

**Models.** We implemented pose estimation models using Lightning Pose (Biderman et al., 2024). We established a strong baseline utilizing a ResNet-50 backbone pretrained on AP-10K (Yu et al., 2021), which outperforms a DeepLabCut baseline (ImageNet-pretrained ResNet-50) on all but the CRIM13 dataset (Fig. 11). Our second baseline is a Vision Transformer (VIT-B/16) pretrained on ImageNet (He et al., 2022) using our own implementation of ViTPose (Xu et al., 2022), enabling assessment of potential improvements when transitioning from convolutional- to transformer-based architectures. Our own VIT-based models utilize this same architecture. In the Appendix we provide additional baselines that use fine-tuned DINO (Caron et al., 2021), DINOv2 (Oquab et al., 2023), and Segment Anything (Kirillov et al., 2023) encoders (which BEAST consistently outperforms;

Fig. 9). For consistency across all model variants, we employ an identical pose estimation head that transforms backbone features into keypoint heatmaps. Given the spatial nature of the task, we use patch embeddings rather than `CLS` tokens in the transformers, and train all models end-to-end.

**Evaluation.** To rigorously evaluate our pose estimation models, we designed a challenging limited-data scenario with only 100 labeled training frames, a realistic constraint for many research settings where extensive annotation is impractical. We measured pixel error between predicted keypoints and ground truth on a test set of novel subjects. For each backbone, we fit three models on different 100-frame subsets. Results are presented as pixel error relative to ensemble standard deviation (e.s.d.) across all seeds and backbones following Biderman et al. (2024), with error curves showing performance at varying difficulty thresholds. Each point corresponds to keypoints with e.s.d. exceeding the threshold value, with the leftmost portion showing error across all keypoints and rightward movement including only increasingly challenging keypoints (those with higher inter-model variability).

**Results.** We find robust improvements in pose estimation quality across all datasets when utilizing BEAST (Fig. 3). The ImageNet-pretrained VIT outperforms the AP-10K-pretrained ResNet-50 on all datasets except the fish, demonstrating the effectiveness of transformers even with limited labels. Notably, pretraining the transformer with the MAE objective on experiment-specific data yields substantial performance gains across all datasets, including the challenging fish dataset. Augmenting MAE with the contrastive objective (BEAST) produces additional performance improvements for some datasets, with particularly pronounced benefits observed in the IBL dataset. However, pretraining only on the contrastive objective leads to significantly worse results, consistent with the different learning objectives: the temporal contrastive loss emphasizes high-level temporal structure, whereas the MAE loss emphasizes low-level, pixel-level features. Consequently, MAE-pretrained representations are better suited for pixel-level prediction tasks like pose estimation, though addition of the contrastive loss in BEAST still provides complementary benefits. We also find BEAST's performance advantages persist when scaling to larger training datasets (Fig. 10).

### 4.3 ACTION SEGMENTATION

Action segmentation classifies discrete behaviors using spatiotemporal video features (Pereira et al., 2020). Similar to pose estimation, a central challenge in animal action segmentation is the lack of large annotated datasets, as behaviors of interest often vary across species and experimental contexts. Many current approaches rely on keypoints (Branson et al., 2009; Kabra et al., 2013; Segalin et al., 2021; Gabriel et al., 2022; Goodwin et al., 2024), requiring an initial labor-intensive and error-prone preprocessing step. Vision transformer embeddings eliminate this preprocessing requirement and provide an attractive alternative if they match or exceed the performance of keypoint-based methods.

**Datasets.** We present results on two datasets (Fig. 4): (1) the "IBL" dataset (IBL et al., 2025a), which contains four behavior classes for the paw nearest the camera; and (2) the Caltech Mouse Social Interactions (CalMS21) dataset (Sun et al., 2021a), a resident–intruder assay of interacting mouse pairs that contains four social behavior classes.

**Models.** We implemented models using two types of embeddings from frozen VIT models: (1) `CLS` tokens and (2) per-patch embeddings. For `CLS` embeddings, we tested both a linear model and a TCN (Lea et al., 2016), enriching the input with inter-frame differences (Blau et al., 2024). We used a sliding window over this feature sequence to predict the action class of the central frame. For patch embeddings, we applied multi-head attention pooling (Lee et al., 2019; Yu et al., 2022; Sun et al., 2024) to integrate information across patches, then concatenated the resulting frame-level embeddings with their inter-frame differences before processing them through a TCN (Fig. 12).

For IBL, we compared against three baseline features: (1) a single paw keypoint, obtained using five pose estimation networks (each trained with 7,000 labeled frames) post-processed with an Ensemble Kalman Smoother (Biderman et al., 2024); (2) principal components of the video frames; and (3) the `CLS` and patch embeddings extracted from a frozen-weight VIT pretrained on ImageNet with MAE loss (which outperformed patch embeddings from DINOv2 in both datasets; Table 15). For CalMS21, we compared against four baseline features: (1) Trajectory Embedding for Behavior Analysis (TREBA) (Sun et al., 2021b), a self-supervised feature extraction method for keypoint trajectories; (2) Simple Behavioral Analysis (SimBA) (Goodwin et al., 2024), which extracts hundreds of hand-crafted features from the keypoint trajectories; (3) principal components of video frames; and (4) the `CLS` and patch embeddings from a frozen-weight VIT pretrained on ImageNet with MAE

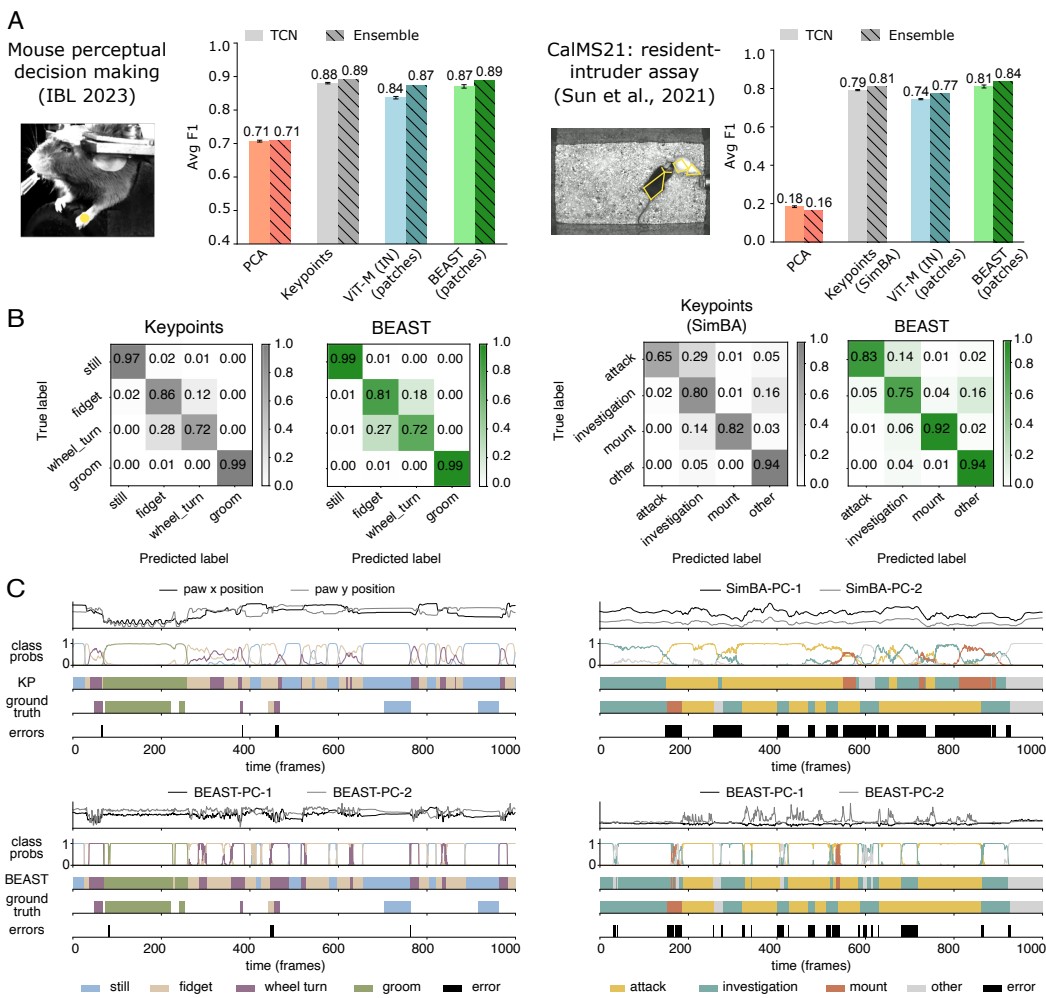

Figure 4: **BEAST improves action segmentation. A:** Example frame from each dataset; performance evaluated across multiple baseline features with both TCN (solid) and ensembled (hatched) models. Error bars represent standard error of the mean across five random initializations. **B:** Confusion matrices for TCN models based on keypoints and BEAST patch embeddings. **C:** Example behavior sequences with feature traces (single seed shown for BEAST models), ensemble probabilities, ensembled model ethograms, ground truth ethograms, and error frames. PCs of SimBA and BEAST features are shown for illustration, but the models utilize the full feature set.

loss. The pose estimator used for TREBA and SimBA was trained with 15,000 labeled frames (Sun et al., 2021a). For all baselines except SimBA, we also concatenated inter-frame differences.

**Evaluation.** All model hyperparameters are tuned to ensure robust baseline performance (Appendix E). We evaluate performance on held-out animals using the macro-averaged F1 score. For the CalMS21 dataset, following Sun et al. (2021a), we average the F1 score over the attack, investigation and mount classes. For all models we train five networks using different random initializations. We also report the results of model ensembles by averaging logits across seeds before applying softmax.

**Results.** BEAST demonstrates strong action segmentation performance across all datasets (Fig. 4). Remarkably, ImageNet-pretrained VIT-M patch embeddings nearly match keypoint-based methods despite utilizing a frozen, general-purpose backbone. This establishes a competitive baseline without requiring the thousands of labels needed to train pose estimation networks. On IBL, BEAST improves upon ImageNet baselines. The keypoint-based model excels here due to action classes corresponding to paw movements easily captured by pose estimation, but this advantage disappears with ensembling: BEAST ensemble F1 matches the keypoint ensemble. CalMS21 better demonstrates BEAST's abilities, which surpasses the SimBA baseline and substantially outperforms the TREBA baseline

(Table 2). The ensembled F1 score of 0.84 places our result in the top 15 of the Multi-Agent Behavior Challenge on AIcrowd.com (top score of 0.89). Additional experiments confirm domain-specific pretraining benefits: BEAST CLS tokens consistently outperform their ImageNet-pretrained counterparts (Table 2), though patch-based models perform significantly better due to their enhanced spatial resolution and multi-headed attention pooling. An ablation experiment on BEAST's loss terms show that backbones pretrained with a contrastive-only (VIT-C) or MAE-only (VIT-M) loss do not perform as well as their combination (Table 15). Across all experiments, nonlinear models consistently outperform their linear counterparts, except for PCA features on CalMS21 (Table 15). All evaluations use frozen backbones with only linear/TCN heads fine-tuned, suggesting further gains may be possible through full backbone fine-tuning.

BEAST's advantages extend beyond absolute F1 improvements. Pose estimation-based approaches require both extensive labeling and iterative training and validation of pose estimation models before action segmentation, often a months- or even years-long process (IBL et al., 2022). BEAST eliminates this entire pipeline, achieving competitive or superior performance using only unlabeled video for pretraining.

Table 2: Action segmentation performance (F1±S.E.M.).

| Method (TCN) | IBL | CalMS21 |
|---|---|---|
| TREBA | – | $0.72 \pm 0.01$ |
| VIT-M (IN) (CLS) | $0.79 \pm 0.00$ | $0.60 \pm 0.00$ |
| VIT-M (IN) (patch) | $0.84 \pm 0.00$ | $0.74 \pm 0.00$ |
| BEAST (IN+PT) (CLS) | $0.81 \pm 0.00$ | $0.63 \pm 0.00$ |
| BEAST (IN+PT) (patch) | $\mathbf{0.87 \pm 0.01}$ | $\mathbf{0.81 \pm 0.01}$ |

## 5 DISCUSSION

This work introduces BEAST, a framework for self-supervised vision transformer pretraining leveraging domain-specific video data. We demonstrated BEAST's significant benefits across neural encoding, pose estimation, and action segmentation tasks. Notably, BEAST outperforms DINOv2 and other computer vision foundation models across all tasks (Table 13), demonstrating the power of domain-specific pretraining. Our frame-based approach is an efficient alternative to native video models like VIDEOMAE (Tong et al., 2022), which require significantly more compute for training and inference (Table 8); however BEAST still outperforms a frozen VIDEOMAE on the neural encoding task (Table 13).

Our work establishes a foundation for several promising future directions. Investigation of transformer attention and learned features could clarify how BEAST operates across different tasks. The black-box nature of ViT embeddings presents interpretability challenges in scientific contexts where transparent representations like pose estimates are often preferred. Visualization methods provide initial insights (Fig. 7), but systematic analysis of what features drive performance on different tasks would strengthen our understanding of when and why BEAST succeeds.

While we have demonstrated BEAST's performance across diverse experimental contexts–including head-fixed and freely moving animals, single- and multi-view setups, and solitary or social behaviors–validation across more environments and species is needed. The success of masked autoencoding and contrastive losses in general computer vision suggests BEAST should adapt well to naturalistic settings (e.g., home cages, zoos, field studies). The primary challenge will be adjusting the frame sampling strategy to accommodate different visual statistics and behavioral distributions in these less controlled environments.

Finally, we see two complementary paths toward making powerful self-supervised models more accessible to individual labs. First, using smaller transformer architectures will reduce pretraining, fine-tuning, and inference costs, but may sacrifice performance. Second, BEAST's framework could enable foundation models of animal behavior trained across diverse datasets, rather than the dataset-specific pretraining we present here. Such foundation models would allow labs to finetune already-powerful pretrained models rather than pretraining themselves. Together, these approaches would lower barriers to adoption and enable wider application of self-supervised learning across the neuroscience community.

**Acknowledgments and Disclosure of Funding**

This work was supported by Gatsby Charitable Foundation GAT3708, NIH U19NS123716 and R50NS145433, NSF 1707398, the National Science Foundation and DoD OUSD (R&E) under Cooperative Agreement PHY-2229929 (The NSF AI Institute for Artificial and Natural Intelligence), Simons Foundation, Wellcome Trust 216324, and funds from Zuckerman Institute Team Science.

## REPRODUCIBILITY STATEMENT

We have attempted to provide sufficient detail to foster reproducibility of all analyses presented in this manuscript.

**Data availability.** All datasets are sourced from public repositories, with links and accompanying licenses provided in Appendix A.

**Model availability.** Pretrained BEAST models for each dataset are available at `https://huggingface.co/collections/paninski-lab/beast` under the CC-BY 4.0 license.

**Code availability.**

- BEAST frame selection, pretraining, and inference: `https://github.com/paninski-lab/beast/releases/tag/v1.0.0`.
- Neural encoding:
  - Reduced Rank Regression for IBL: `https://github.com/realwsq/brainwide-RRR-encoding-model`
  - Reduced Rank Regression for Facemap: `https://github.com/MouseLand/facemap/blob/v1.0.7/facemap/neural_prediction/prediction_utils.py#L110`
  - Temporal Convolution Network: `https://github.com/MouseLand/facemap/blob/v1.0.7/facemap/neural_prediction/neural_model.py`
- Pose estimation: `https://github.com/paninski-lab/lightning-pose/releases/tag/v2.0.2`
- Action segmentation: `https://github.com/paninski-lab/lightning-action/releases/tag/v1.0.1`.

**Analysis details.**

- Pretraining (Appendix B): BEAST pretraining procedure and frame selection strategy.
- Neural encoding (Appendix C): Model architectures, training procedures, and hyperparameter tuning details.
- Pose estimation (Appendix D): Training procedures and hyperparameter selection.
- Action segmentation (Appendix E): Model architectures, training procedures, and hyperparameter tuning details.

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

# Appendix

## A  DATASETS

All datasets used for this study were collected in compliance with the relevant ethical regulations (see the references for each dataset).

### A.1  IBL

This dataset (IBL, 2023) from the International Brain Lab (IBL) and consists of head-fixed mice performing a decision-making task (IBL et al., 2021; 2025b;a). Two cameras–'left' (60 Hz) and 'right' (150 Hz)–capture roughly orthogonal side views of the mouse's face and upper trunk during each session. Frames are downsampled to $256 \times 320$ pixels for labeling and video storage. We accessed the raw videos and neural activity under the CC-BY 4.0 license using these instructions: `https://int-brain-lab.github.io/iblenv/notebooks_external/data_release_brainwidemap.html`. Frames for pretraining are available at `https://doi.org/10.6084/m9.figshare.31310143` under the CC-BY 4.0 license.

**Pose estimation.** Frames were reshaped during training to $256 \times 256$ pixels. Two keypoints were labeled per view, one for each paw. We accessed the initial pose estimation labels from the public repository at `https://ibl-brain-wide-map-public.s3.amazonaws.com/aggregates/Tags/2023_Q1_Biderman_Whiteway_et_al/_ibl_videoTracking.trainingDataPaw.7e79e865-f2fc-4709-b203-77dbdac6461f.zip` under the CC-BY 4.0 license. This dataset contains 6,071 labeled train frames from 35 animals and 1,446 labeled test frames from 10 animals, all at $128 \times 102$ pixel resolution. Despite the large number of labeled frames, we observed poor performance in sessions with bright lights or other distractors. Additionally, the low resolution often obscured fine details, for example complicating discrimination of individual paws when close together. To address these limitations, we retrieved the full-resolution frames ($1280 \times 1024$ for left view, $640 \times 512$ for right) from the raw videos and downsampled them to $320 \times 256$ pixels. This higher resolution revealed occasional labeling errors, which we manually corrected. We then added 1,437 newly labeled frames from 15 additional animals, creating an expanded training set of 7,609 frames from 50 animals. The updated pose estimation dataset is available at `https://ibl-brain-wide-map-public.s3.amazonaws.com/aggregates/Tags/2026_Q1_Wang_Yu_et_al/_ibl_videoTracking.trainingDataPaw.zip` under the CC-BY 4.0 license.

**Action segmentation.** Four action classes are labeled for the paw closest to the camera: (1) still; (2) grooming; (3) turning the wheel; and (4) fidget (any movement that is not grooming or wheel turning). We accessed the initial action segmentation labels from `https://doi.org/10.6084/m9.figshare.27479760.v1` under the CC-BY 4.0 license. This dataset contains 1,000,000 frames from 10 animals, of which 14,107 are labeled.

We expanded this dataset as the original study (Blau et al., 2024) only used five animals each for training and testing. First, we trained an ensemble of five TCN-based action segmentation models on all 10 existing animals. We applied these models to a new batch of 53 sessions and calculated the variance in predicted probabilities across all models for each frame. The 19 sessions with the highest average ensemble variance (indicating where models disagreed most) were selected for further labeling. We then labeled 36,009 additional frames from these sessions. For the analyses in this paper, we selected the subset of all labeled sessions that are included in the BEAST pretraining sessions, and split these into train (32,521 frames from 18 animals) and test sets (7,786 frames from 5 animals). The updated action segmentation dataset is available at `https://ibl-brain-wide-map-public.s3.amazonaws.com/aggregates/Tags/2026_Q1_Wang_Yu_et_al/_ibl_videoActionSegmentation.trainingDataNearPaw.zip` under CC-BY 4.0 license.

**Neural encoding.** For the neural analysis we use a subset of the IBL repeated site dataset (IBL et al., 2025b). This dataset consists of Neuropixels recordings collected from 10 labs with standardized experimental pipelines. The recordings target the same five brain regions across all mice: VISa (primary visual cortex), CA1 and DG (hippocampus), and LP and PO (thalamic nuclei). We evaluate neural encoding models on five randomly selected sessions. Moreover, we used the trial-aligned

neural activity data, taking 2 seconds of activity aligned to wheel movement onset. We binned the spikes every 20 ms to get a total of 100 bins per trial. We also filtered out the low firing rate neurons by setting a minimum threshold of 2 Hz. For each trial we randomly select 100 video frames (out of a possible 120) to fit the downstream neural encoding models, resulting in an effective sampling rate of 50 Hz to match the neural data.

## A.2 IBL-WHISKER

We localize the whisker pad using anchor keypoints on the nose and eye, following the procedure in IBL et al. (2022). We use the same sessions and neural activity as the "IBL" dataset.

## A.3 MIRROR-MOUSE

Head-fixed mice ran on a circular treadmill while avoiding a moving obstacle (Warren et al., 2021). The treadmill had a transparent floor and a mirror mounted inside at 45°, allowing a single camera to capture two roughly orthogonal views (side view and bottom view via the mirror) at 250 Hz. The camera was positioned at a large distance from the subject (∼1.1 m) to minimize perspective distortion. Frames are $406 \times 396$ pixels and reshaped during pose estimation training to $256 \times 256$ pixels. Seventeen keypoints were labeled across the two views including seven keypoints on the mouse's body per view, plus three keypoints on the moving obstacle. The full training dataset consists of 789 labeled frames across 10 animals; the test dataset consists of 253 labeled frames across three animals. We accessed the labeled pose estimation dataset from `https://doi.org/10.6084/m9.figshare.24993315.v1` under the CC-BY 4.0 license. Frames for pretraining are available at `https://doi.org/10.6084/m9.figshare.31320529` under the CC-BY 4.0 license.

## A.4 MIRROR-FISH

Mormyrid fish of the species Gnathonemus petersii swam freely in and out of an experimental tank, capturing worms from a well (Biderman et al., 2024; Pedraja et al., 2025). The tank had a side mirror and a top mirror, both at 45°, providing three different views seen from a single camera at 300 Hz. The camera was placed ∼1.7 m away from the center of the fish tank to reduce distortions. Frames are $384 \times 512$ pixels and reshaped during training to $256 \times 384$ pixels. Seventeen body parts were labeled across each of three views for a total of 51 keypoints. The full training dataset consists of 373 frames across three animals; the test dataset consists of 94 frames across three animals. We accessed the labeled pose estimation dataset from `https://doi.org/10.6084/m9.figshare.24993363.v1` under the CC-BY 4.0 license. Frames for pretraining are available at `https://doi.org/10.6084/m9.figshare.31320775` under the CC-BY 4.0 license.

## A.5 CRIM13

The Caltech Resident-Intruder Mouse (CRIM13) dataset (Burgos-Artizzu et al., 2012) consists of two mice interacting in an enclosed arena, captured by a top-view camera at 30 Hz. Frames are $480 \times 640$ pixels and reshaped during pose estimation training to $256 \times 256$ pixels. Seven keypoints were labeled on each mouse for a total of 14 keypoints (Segalin et al., 2021). The full training dataset consists of 3,986 frames across four resident mice; the test dataset consists of 1,274 frames across the same four resident mice but a different set of intruder mice. The original dataset is available at `https://data.caltech.edu/records/4emt5-b0t10`. We accessed the labeled pose estimation dataset from `https://doi.org/10.6084/m9.figshare.24993384.v1` under the CC-BY 4.0 license. Frames for pretraining are available at `https://doi.org/10.6084/m9.figshare.31321213` under the CC-BY 4.0 license.

## A.6 CALMS21

The Caltech Mouse Social Interactions (CalMS21) dataset (Sun et al., 2021a), like CRIM13, consists of two mice interacting in an enclosed arena, captured by a top-view camera at 30 Hz. The dataset consists of videos with tracked poses and frame-level behavior annotations. Four behavior classes are labeled: attack, investigation, mount, and other (i.e., none of the above). The full training dataset consists of 506,668 frames across 68 videos; the test dataset consists of 262,107 frames across 19 videos. We accessed the pose estimates, TREBA features (Sun et al., 2021b), and behavior annotations from `https://doi.org/10.22002/D1.1991` under the CC-BY 4.0 license. Frames for pretraining are available at `https://doi.org/10.6084/m9.figshare.31337953` under the CC-BY 4.0 license.

### A.7 FACEMAP

Head-fixed mice were free to run on an air-floating ball in darkness (Syeda et al., 2024). A single infrared camera captured one of several side or front views of the mouse's face and upper trunk during each session at 50 Hz. Fifteen keypoints were labeled across the face (mouse, nose, whiskers, eyes) and paw. Neural activity was recorded across visual and sensorimotor areas using two-photon calcium imaging at 3 Hz. Approximately 30,000 to 50,000 cells were recorded in a given session. In our encoding task, we predict the 128 neural principal components following Syeda et al. (2024). We evaluate neural encoding models on five randomly selected sessions. The publicly available data did not contain additional videos, so we only fine-tuned neural encoding models with this dataset. We accessed the raw videos, pose estimates, and neural activity from `https://doi.org/10.25378/janelia.23712957` under the CC-BY-NC 4.0 license.

Table 3: Number of training/validation/test frames utilized across tasks, with number of source videos in parentheses. Pretraining frames are unlabeled, and the number includes both anchor frames and temporal context frames. Pose estimation and action segmentation frames are labeled; these models are trained across multiple videos. Neural encoding frames have matched neural activity (output) for each time point of behavior (input); these models are trained on single videos, since the neural populations change from one session to the next.

| | Pretraining | Pose estimation | | Action segmentation | | Neural encoding | |
|---|---|---|---|---|---|---|---|
| | | train/val | test | train/val | test | train/val | test |
| IBL (IBL et al., 2025a) | 270,025 (77) | 7,609 (128) | 1,446 (19) | 35,521 (18) | 7,786 (5) | 338,760 (5) | 42,720 (5) |
| Mirror-mouse (Warren et al., 2021) | 94,272 (17) | 789 (17) | 253 (5) | - | - | - | - |
| Mirror-fish (Biderman et al., 2024) | 47,943 (28) | 373 (28) | 94 (10) | - | - | - | - |
| CRIM13 (Burgos-Artizzu et al., 2012) | 99,926 (36) | 3,986 (37) | 1,274 (19) | - | - | - | - |
| CalMS21 (Sun et al., 2021a) | 91,087 (70) | - | - | 506,668 (68) | 262,107 (19) | - | - |
| Facemap (Syeda et al., 2024) | - | - | - | - | - | 1,790,200 (5) | 447,550 (5) |

## B BEAST IMPLEMENTATION

BEAST utilizes a standard VIT-B/16 architecture (Dosovitskiy et al., 2020), and combines a masked autoencoding and temporal contrastive learning loss. This approach is also taken by VIC-MAE (Hernandez et al., 2024), and we introduce key adaptations and simplifications to make BEAST suitable for applications in behavioral neuroscience, which we elaborate on more in the following sections.

### B.1 ARCHITECTURE

We selected the "base" VIT-B/16 architecture over other VIT variants because it is expressive enough to capture rich frame-level information while remaining computationally efficient for training and inference on long videos. While VIC-MAE employs a pooled attention layer to transform patch embeddings into a 768-dimensional vector that serves as the global image representation for the temporal contrastive loss, we use the standard CLS token (Table 7). This CLS token is passed through a nonlinear projector consisting of four components: a linear layer, Batch Norm (necessary for stable training, Fig. 5), ReLU activation, and a final linear layer.

The BEAST backbone is initialized using weights pretrained on ImageNet with a masked autoencoding loss. This model has exceptional zero-shot performance on neural encoding, outperforming all other baselines, and is further improved with domain-specific pretraining (Table 1). We find other pretrained backbones–DINOv2 (Oquab et al., 2023) and CLIP (Radford et al., 2021)–also have strong zero-shot performance, but do not significantly improve upon MAE+ImageNet (Table 13), indicating this is a reasonable pretrained backbone from which to start our own domain-specific pretraining (see Fig. 9 and Table 15 for similar results on pose estimation and action segmentation, respectively).

BEAST incorporates time through its temporal contrastive loss, which efficiently captures information across frames. We explored the performance of VIDEOMAE (Tong et al., 2022), a related video model pretrained on Kinetics-400 (Kay et al., 2017). We found that the frozen VIDEOMAE backbone does outperform the frame-based VIT-MAE on the neural encoding task (Table 13). Interestingly BEAST, which applies additional domain-specific pretraining to VIT-MAE, still outperforms VIDEOMAE. This raises the intriguing question of whether further domain-specific pretraining of VIDEOMAE could surpass BEAST performance. We view this as an important direction for future work.

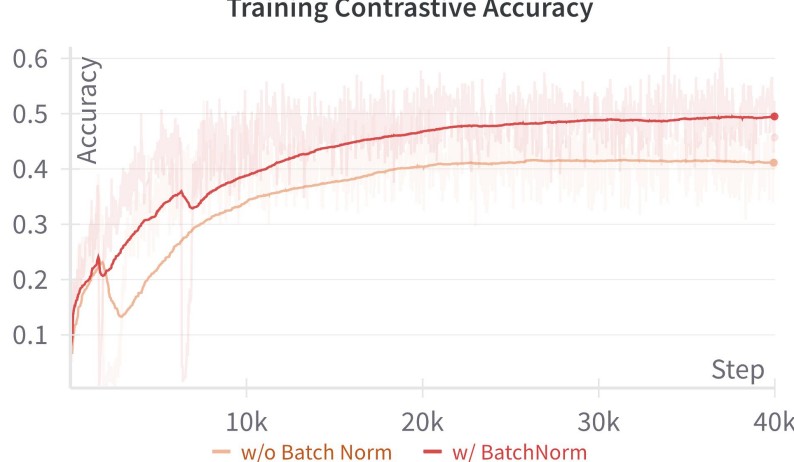

Figure 5: **Effect of Batch Normalization on contrastive training accuracy.** Training contrastive accuracy improves significantly with the use of Batch Normalization (BatchNorm) in the nonlinear projection head. Models trained with BatchNorm exhibit smoother learning curves and achieve higher final accuracy compared to those without BatchNorm. "Accuracy" is defined as the fraction of anchor frames in a batch where the corresponding positive frame has a logit score higher than that of all other negative frames.

## B.2 TRAINING

We discuss frame selection and sampling strategies, data augmentations, and global pooling strategies below. We apply the MAE loss uniformly across all frame types (anchor, positive, negative), whereas VIC-MAE only applies the MAE loss to anchor frames. The global batch size is set to 2048, distributed across 8 Nvidia A40 GPUs. We use the AdamW optimizer with a weight decay of 0.05. The learning rate is scheduled using PyTorch's `OneCycleLR` scheduler, with a base learning rate of $5 \times 10^{-5}$. The maximum learning rate is computed as $\text{max\_lr} = \text{base\_lr} \times \frac{\text{global\_batch\_size}}{256}$, with `pct_start` set to 0.15 and `div_factor` set to 10. We train all models for 800 epochs.

**Frame selection strategy.** Animal behavior videos often contain extended periods of inactivity or repetitive behaviors. Pretraining VIT models on all available frames would capture redundant information and increase computation time. Instead, we extract diverse frames exhibiting distinct poses (to optimize the masked autoencoding loss) while preserving local temporal relationships (to leverage the temporal contrastive loss). We first downsample all frames to $32 \times 32$ pixels and calculate motion energy, defined as the absolute pixel-wise differences between consecutive frames. We eliminate frames in the bottom 50th percentile of motion energy, retaining only those with significant movement. We then apply $k$-means clustering to the remaining downsampled frames, with the number of clusters matching our target number of anchor frames per video (e.g., 600). For each cluster, we select the frame closest to the cluster center, along with its immediate predecessor and successor in time, which serve as positive examples for the contrastive loss (for a total of, e.g., 1800 frames per video). This methodology ensures a high-quality, diverse dataset for efficient pretraining and outperforms random frame selection in the neural encoding task (Table 4).

Table 4: Frame selection strategy ablation. We pretrain models using either random selection ("Random") or the PCA+$k$-means strategy described above ("Selected"). We evaluate the representations of these models using zero-shot performance on the neural encoding task using the bits per spike (BPS) metric. We report the mean and standard deviation of BPS across five test sessions.

| Method | IBL | | IBL-whisker | |
|---|---|---|---|---|
| | TCN | RRR | TCN | RRR |
| VIT-M (Random) | $0.311 \pm 0.107$ | $0.168 \pm 0.091$ | $0.319 \pm 0.084$ | $\mathbf{0.147 \pm 0.026}$ |
| BEAST (Random) | $0.319 \pm 0.104$ | $0.176 \pm 0.094$ | $\mathbf{0.319 \pm 0.081}$ | $0.138 \pm 0.035$ |
| BEAST (Selected) | $\mathbf{0.337 \pm 0.103}$ | $\mathbf{0.177 \pm 0.076}$ | $0.317 \pm 0.083$ | $0.138 \pm 0.029$ |

**Frame sampling strategy during training.** Once we have a diverse set of training frames, we must construct batches during training. As stated in Sec. 3, for a batch of size $B$ we randomly select $B/2$ anchor frames, which can originate from any and all videos. For BEAST, each anchor frame $\mathbf{x}_t^v$ is paired with a positive frame randomly selected from $\mathbf{x}_{t\pm1}^v$. All other frames serve as negative frames, including frames from the same video. Due to the frame selection strategy described above, even frames from the same video will be visually distinct and not interfere with the contrastive loss (Fig. 6). This batch construction procedure is distinct from VIC-MAE, which allows any two frames from the same video to be a positive pair, while only frames from different videos are negative pairs. We find our approach outperforms the VIC-MAE approach in the neural encoding task (Table 5). This sampling strategy only applies to BEAST; the VIT-M models do not contain the contrastive loss, and we only train them with the anchor frames.

Table 5: Frame sampling strategy ablation. We pretrain models using either the VIC-MAE or BEAST frame *sampling* strategy; both models use the superior "Selected" frame *selection* strategy. We evaluate the representations of these models using zero-shot performance on the neural encoding task using the bits per spike (BPS) metric. We report the mean and standard deviation of BPS across five test sessions.

| Method | IBL | | IBL-whisker | |
| --- | --- | --- | --- | --- |
| | TCN | RRR | TCN | RRR |
| ViC-MAE (IN+PT) | $0.331 \pm 0.103$ | $0.141 \pm 0.080$ | $0.289 \pm 0.055$ | $0.127 \pm 0.033$ |
| BEAST (IN+PT) | $\mathbf{0.337 \pm 0.103}$ | $\mathbf{0.177 \pm 0.076}$ | $\mathbf{0.317 \pm 0.083}$ | $\mathbf{0.138 \pm 0.029}$ |

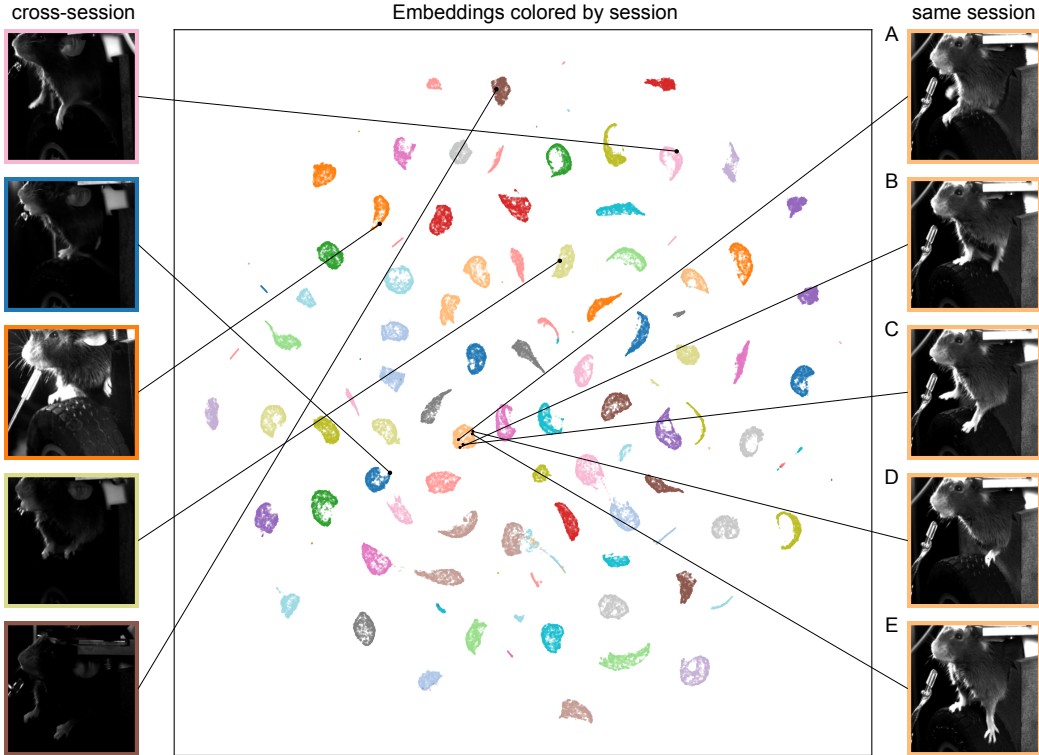

Figure 6: **Temporal contrastive loss visualization.** UMAP embeddings (McInnes et al., 2018) of the first 32 principal components (PCs) of all anchor frames used for pretraining on the IBL dataset. Each anchor frame is colored by the session it was sampled from. The left and right columns show example frames drawn from the same session and from different sessions, respectively. The sampled frames are largely visually distinct, even frames next to each other in UMAP space (e.g., frames B and C, or D and E, on the left-hand side).

**Data augmentation.** The default data augmentation procedure (He et al., 2022) applies a random resized crop to $244 \times 244$ pixels with a crop ratio between 0.2 and 1.0, followed by a random

horizontal flip with probability 50%. We explored an extended augmentation strategy that adds further transformations: rotation up to 45 degrees and color jittering (brightness=0.4, contrast=0.4, saturation=0.4, hue=0.1), in addition to the crop and flip. We compare the performance of BEAST using both the default and the extended augmentation strategy. The additional augmentations achieve performance similar to the default setting (Table 6), so we use the default throughout the paper.

Table 6: Data augmentation ablation. We pretrain models using either default or extended data augmentations. We evaluate the representations of these models using zero-shot performance on the neural encoding task using the bits per spike (BPS) metric. We report the mean and standard deviation of BPS across five test sessions.

| Method | IBL | | IBL-whisker | |
| --- | --- | --- | --- | --- |
| | TCN | RRR | TCN | RRR |
| BEAST (default data aug) | $\mathbf{0.337 \pm 0.103}$ | $\mathbf{0.177 \pm 0.076}$ | $\mathbf{0.317 \pm 0.083}$ | $0.138 \pm 0.029$ |
| BEAST (extend data aug) | $0.328 \pm 0.102$ | $0.163 \pm 0.081$ | $0.314 \pm 0.077$ | $\mathbf{0.150 \pm 0.042}$ |

**Pooling strategy.** The `CLS` token serves as a global frame representation, effectively pooling information across all spatial positions into a single latent vector. To explore alternative pooling strategies during pretraining, we conducted additional experiments by pretraining models using mean pooling and attention pooling of the patch embeddings, then evaluating performance on the neural encoding task (Table 7). The ablation results clearly demonstrate that the CLS token is the most effective aggregation method for pretraining.

Table 7: Pooling ablation. We pretrain models using either the `CLS` token, or mean or attention pooling of the patch embeddings, to aggregate information across the image for the temporal contrastive loss. We evaluate the representations of these models using zero-shot performance on the neural encoding task using the bits per spike (BPS) metric. We report the mean and standard deviation of BPS across five test sessions.

| Method | IBL | | IBL-whisker | |
| --- | --- | --- | --- | --- |
| | TCN | RRR | TCN | RRR |
| BEAST (mean pooling) | $0.321 \pm 0.104$ | $0.159 \pm 0.080$ | $0.302 \pm 0.082$ | $0.128 \pm 0.025$ |
| BEAST (attention pooling) | $0.323 \pm 0.100$ | $0.141 \pm 0.076$ | $0.307 \pm 0.072$ | $0.130 \pm 0.031$ |
| BEAST (CLS token) | $\mathbf{0.337 \pm 0.103}$ | $\mathbf{0.277 \pm 0.076}$ | $\mathbf{0.317 \pm 0.083}$ | $\mathbf{0.138 \pm 0.029}$ |

### B.3 HYPERPARAMETER DETAILS

The BEAST objective combines reconstruction and contrastive losses. During the early stages of training, it is important for the model to focus on accurately reconstructing the input, so the reconstruction loss should dominate. As the model learns to capture low-level pixel structure, the contrastive loss gradually becomes more important. It acts as a regularizer, encouraging the model to learn higher-level temporal representations rather than overfitting to local pixel patterns.

The weighting factor for the contrastive loss, $\lambda$, plays a crucial role during pretraining; too large and the model will not reconstruct the input well; too small and the model does not reap its regularizing benefits. Through hyperparameter tuning based on neural encoding performance (on the validation set), we set $\lambda = 0.01$ for all datasets, except IBL and mirror-mouse, where we set $\lambda = 0.02$. This choice ensures the reconstruction loss is emphasized in the early phases of training, while the contrastive loss naturally takes over as reconstruction error decreases toward the end, eliminating the need for an annealing schedule.

The masking strategy for the reconstruction loss also significantly impacts performance. We found that an aggressive mask ratio of 0.75 works effectively across various tasks, for both ViT-M and BEAST. When fine-tuning BEAST for neural encoding models (IN+FT or IN+PT+FT), we tested mask ratios of 0.75 and 0.9, with 0.9 performing better on validation data. These 0.9-ratio models are used for the final fine-tuning results presented in our tables and figures.

### B.4 COMPUTATIONAL EFFICIENCY COMPARISON

To compare the computational efficiency of image (ViT-M, BEAST) versus video models (VIDEOMAE), we recorded three metrics for each model: runtime (ms per batch), Giga Float-

ing Point Operations per Second (GFLOPS), and memory required for a forward pass with batch size one. All experiments were run using the `fvcore` library on a single A100 GPU. Since runtime varies across batches while GFLOPS and memory are deterministic, we report mean and standard deviation of runtime across 32 batches. We benchmark two modes: "pretrain", which uses patch masking (0.75 for image models, 0.9 for VideoMAE, following optimal settings from the respective papers); and "finetune", which omits patch masking as relevant for our downstream tasks.

BEAST and VIT-M show comparable performance across all metrics, with BEAST having a slightly longer runtime and larger memory footprint due to the nonlinear projector used in the contrastive loss (not substantial enough to affect GFLOPS). VIDEOMAE requires substantially more resources due to processing 16 consecutive frames per batch element: during pretraining, it requires 2.5× runtime, 3.5× GFLOPS, and >2× memory compared to BEAST. These differences are even more pronounced during finetuning when patches are not masked: VIDEOMAE requires 11× runtime, 11× GFLOPS, and 7× memory.

Table 8: Computational efficiency comparison. We benchmark runtime (mean and standard deviation across 32 batches), GFLOPS, and memory usage for BEAST, VIT-M, and VIDEOMAE during pretraining (with patch masking) and finetuning (without patch masking). All measurements are for a forward pass with batch size of one on a single A100 GPU.

| Model | Runtime (ms/batch) | | GFLOPS | | Memory (MB) | |
|---|---|---|---|---|---|---|
| | Finetune | Pretrain | Finetune | Pretrain | Finetune | Pretrain |
| ViT-M | 5.43 ± 0.70 | 7.6 ± 2.43 | 17.59 | 9.80 | 369.0 | 332.72 |
| BEAST | 6.21 ± 0.94 | 7.9 ± 2.26 | 17.59 | 9.80 | 409.1 | 333.40 |
| VideoMAE | 71.51 ± 3.50 | 20.16 ± 2.47 | 199.49 | 35.24 | 2955.1 | 831.61 |

## B.5  FEATURE VISUALIZATIONS

We computed PCA on patch embeddings from 100 randomly sampled frames in the test set and visualized the first three components as RGB channels of two example frames. Compared to DINOv2–whose embeddings emphasize broader semantic structures–the BEAST pre-trained model captures finer-grained details that are critical for neural encoding, pose estimation and behavior segmentation. In contrast, the ViT-M model pretrained only on ImageNet produces patch embeddings that appear noisier and less structurally coherent.

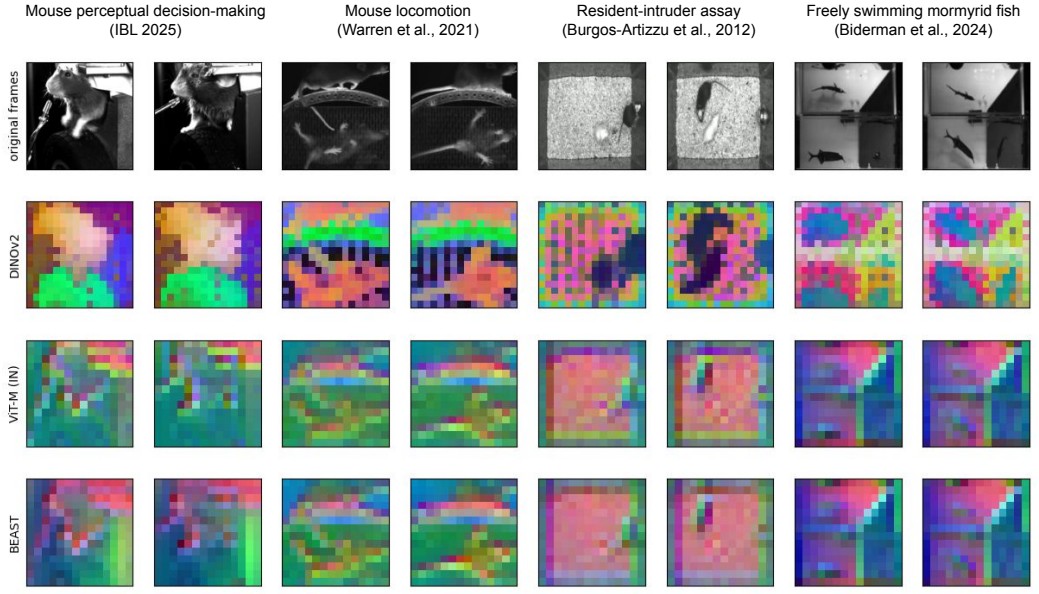

Figure 7: **Visualization of the first three PCA components across models.**

# C    NEURAL ENCODING

## C.1    FEATURE REPRESENTATIONS

**Keypoints.** For the IBL dataset we use 11 keypoints in the publicly available dataset: left and right paws, two edges of the tongue, two edges of the lick spout, nose, and four edges of the pupil. For the Facemap dataset we use 12 keypoints in the publicly available dataset: three whiskers, four points on the nose, four corners of the eye, and the one visible paw.

**Whisker pad motion energy.** We localize the whisker pad in the IBL dataset using anchor keypoints on the nose and eye. We then compute the motion energy of the whisker pad as the absolute pixel-wise differences between consecutive frames, resulting in a one-dimensional representation at each time point (IBL et al., 2022).

**Principal Component Analysis.** We compute PCA on a per-session basis using all frames in the video. A subset of the resulting PCs are used for neural encoding. See Sec. C.6 for information on our dimensionality ablation experiment.

**CEBRA.** CEBRA (Schneider et al., 2023) is a contrastive learning approach that provides a baseline which is complementary to the VIT-M models pretrained on ImageNet with a masked autoencoding objective. Similar to our PCA approach, we train an individual unsupervised CEBRA model for each session using the convolutional neural network option and the default `offset10-model`.

**DINOv2.** For the DINOv2 model (Oquab et al., 2023), we extract the `CLS` embedding for each frame using a frozen pretrained backbone, and use these as input to the encoding models. We did not pretrain this model ourselves, but rather used the model checkpoint available at `https://huggingface.co/facebook/dinov2-base`.

**ViT-M variants.** For the VIT-M models, we extract the `CLS` embedding for each frame using a frozen pretrained backbone, and use these as input to the encoding models. Alternative approaches could include: (1) using patch embeddings with a multi-head attention pooling layer (as in our action segmentation work), or (2) fine-tuning the backbone itself while using either `CLS` or patch embeddings (similar to our approach for pose estimation). We expect these alternative approaches would improve performance and plan to explore them in future work.

- VIT-M (IN): VIT-M model pretrained on ImageNet using a masked autoencoding loss. We did not pretrain this model ourselves, but rather used the model checkpoint available at `https://huggingface.co/facebook/vit-mae-base`.
- VIT-M (IN+PT): Initialized with the ImageNet-pretrained weights, then further pretrained on dataset-specific frames.
- VIT-M (IN+PT+FT): Initialized with the dataset-specific pretrained weights (IN+PT), and then further fine-tuned on a single session.

All training (except VIT-M (IN)) is performed as described in Appendix B.

**ViT-C variants.** For the VIT-C (IN+PT) model, we start with the VIT-M (IN) weights then continue pretraining on dataset-specific frames with the temporal contrastive loss only. Therefore there is no VIT-C (IN) variant. We did not perform additional session-specific fine-tuning, so there is no VIT-C (IN+PT+FT) variant.

**BEAST variants.** The BEAST model variants follow the same naming pattern as VIT-M, with one exception: there is no "BEAST (IN)" variant, as BEAST requires video frames rather than just static images for pretraining.

## C.2    REDUCED RANK REGRESSION

**IBL.** For the IBL dataset, we followed the Reduced Rank Regression (RRR) setup described in (Posani et al., 2025). We trained all models using the L-BFGS optimizer and set the rank constraint to 3. To denoise the neural signals, we applied a 1-dimensional smoothing filter to the neural activity. The hyperparameter search (Sec. C.4) was conducted over the ranges specified in Table 9.

**Facemap.** For the Facemap dataset, we followed the setup described in (Syeda et al., 2024), using the implementation provided in the official Facemap repository. To deal with different neural and

Table 9: RRR model hyperparameters for IBL dataset.

| Hyperparameter | Value Range |
| --- | --- |
| Output Dim | Number of Neurons |
| Rank | 3 |
| Optimizer | L-BFGS |
| Learning Rate | Log-Uniform(0.1, 2) |
| L2 | 100 |

behavioral sampling rates, this model first resamples the behavioral timestamps to match the neural timestamps, and then fits a Reduced Rank Regression model using low-rank SVD. We adopted the default rank of 32 as used in the original implementation. The Lambda parameter refers to the regularization strength, which we set to a relatively low value to avoid over-penalizing the weights. The output dimensionality was set to 128, corresponding to the number of neural principal components used in the model. Model parameters are estimated via a closed-form least squares approach. The hyperparameters we used are specified in Table 10.

Table 10: RRR model hyperparameters for Facemap dataset.

| Hyperparameter | Value Range |
| --- | --- |
| Output Dim | 128 |
| Rank | 32 |
| Lambda | 1e-6 |

### C.3 TEMPORAL CONVOLUTION NETWORK

We used the same implementation of the Temporal Convolution Network (TCN) to process frame embeddings for both the IBL and Facemap datasets, based on the official Facemap repository. The convolutional kernel operates along the temporal dimension of the input (behavioral) data. To deal with different neural and behavioral sampling rates, this model resamples the resulting latent representation at neural timestamps using nearest-neighbor indexing. The TCN model was trained for 300 epochs using the AdamW optimizer, with learning rate decimation (multiplied by 0.1) applied at epochs 120 and 200. The hyperparameter search (Sec. C.4) was conducted over the ranges specified in Table 11.

Table 11: TCN model hyperparameters; [*]IBL dataset; [**]Facemap dataset

| Hyperparameter | Value Range |
| --- | --- |
| Output Dim | Number of Neurons[*], 128[**] |
| Learning Rate | Log-Uniform(5e-5, 2e-3) |
| Optimizer | AdamW |
| Weight Decay | 1e-4 |

### C.4 HYPERPARAMETER SELECTION

For the IBL data, we divided the trials into train (80%), validation (10%), and test (10%) sets. For the Facemap data we followed the experimental setup as described in the original study (Syeda et al., 2024): the session is split into ten blocks; the first 75% of each block is assigned to the training set; the following 3 seconds are excluded to remove data leakage due to autocorrelation in behavior and neural activity; and the final set of frames from the block are assigned to the test set. There is no validation set. For both the RRR and TCN models, we conducted hyperparameter searches separately for each feature type to identify the best-performing configurations. Specifically, we performed 30 runs of randomly selected hyperparameters per model type, evaluating performance on the validation set (IBL) or test set (Facemap) using an evaluation metric specific to each dataset: bits per spike (BPS) for IBL and variance explained for Facemap. The only exception was the RRR model for Facemap, where the parameters were fixed according to the original implementation. We select the

model with the best performance on the validation (IBL) or test (Facemap) set, and report results on the test set.

## C.5 PATCH EMBEDDINGS

The output of the VIT encoder consists of a `CLS` token embedding $z_{\text{CLS}} \in \mathbb{R}^D$ and patch token embeddings $z \in \mathbb{R}^{N \times D}$, where $D = 768$ is the embedding dimension and $N$ is the total number of image patches ($N = 196$ for a $224 \times 224$ input frame). We compare the `CLS` token and attention-pooled patch embeddings as inputs for neural encoding (see implementation details in E.4), and find that the `CLS` token outperforms the patch embeddings (Table 12). Given its lower dimensionality and superior results, we adopt the `CLS` token representation for all subsequent encoding tasks.

Table 12: Comparison of `CLS` and patch embeddings with a TCN encoder. We report the mean and standard deviation of BPS across five test sessions.

| Method | IBL | IBL-whisker |
|---|---|---|
| BEAST (CLS) | $\mathbf{0.337 \pm 0.103}$ | $\mathbf{0.317 \pm 0.083}$ |
| BEAST (patch) | $0.278 \pm 0.065$ | $0.288 \pm 0.070$ |

## C.6 DIMENSIONALITY ABLATION EXPERIMENTS

One of the most important hyperparameters for the PCA, CEBRA, and VIT models is the latent/embedding dimensionality. To thoroughly explore performance across this parameter, we tested these models using various dimensionality values. For a given dimensionality $k$, we used different approaches: (1) for PCA, we selected the top $k$ principal components; (2) for CEBRA, we retrained the model with $k$ latent dimensions; and (3) for VIT models, due to computational constraints, we first trained the full 768-dimensional models, then applied PCA to the embedding space and selected the top $k$ VIT principal components. For each feature, model type, and dimensionality $k$, we fit downstream neural encoding models using the complete hyperparameter search described previously. Figure 2 reports the best result for each model, though notably BEAST outperforms all baselines across all dimensionality values (Fig. 8).

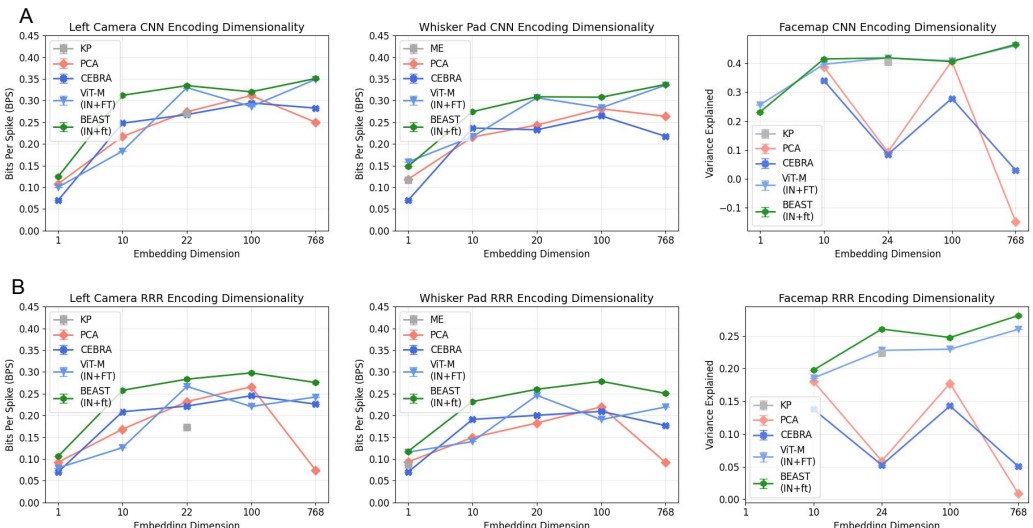

Figure 8: **Encoding performance as a function of embedding dimension.** BEAST outperforms all other baselines over embedding dimensions spanning several orders of magnitude, demonstrating the superiority of its representations for any given dimensionality. Results for Keypoints and Motion Energy are included at their respective dimensionalities for each dataset. The Facemap encoding results for 1-dimensional data performed poorly and included NaN values in some sessions, so we excluded them from the figure.

## C.7 Extended neural encoding results

We collect all neural encoding results in Table 13. The values for PCA and CEBRA correspond to the 100-dimensional results in Fig. 8; the values for VIT-M and BEAST correspond to the full 768-dimensional models.

Table 13: Neural encoding results across feature types and models. All VIT-based models use CLS tokens from a frozen backbone. "IN" refers to a model pretrained with ImageNet weights; "IN+PT" refers to models that are initialized with ImageNet-pretrained weights then further pretrained on experiment-specific data; "+FT" refers to models that are initialized with pretrained weights based on what comes before "+" then fine-tuned on individual sessions. For IBL and IBL-whisker we report the mean and standard error of the mean of BPS over $N$=842 neurons across five test sessions. For Facemap we report the mean and standard deviation of Variance Explained of the first 128 Principal Components of neural activity across five test sessions.

| Features | IBL (BPS) | | IBL-whisker (BPS) | | Facemap (Var Exp) | |
|---|---|---|---|---|---|---|
| | RRR | TCN | RRR | TCN | RRR | TCN |
| Keypoints | $0.169 \pm 0.008$ | $0.269 \pm 0.011$ | – | – | $0.224 \pm 0.047$ | $0.403 \pm 0.077$ |
| Motion energy | – | – | $0.086 \pm 0.006$ | $0.113 \pm 0.007$ | – | – |
| PCA | $0.260 \pm 0.010$ | $0.309 \pm 0.012$ | $0.212 \pm 0.009$ | $0.272 \pm 0.011$ | $0.177 \pm 0.064$ | $0.407 \pm 0.090$ |
| CEBRA | $0.239 \pm 0.010$ | $0.293 \pm 0.012$ | $0.204 \pm 0.009$ | $0.260 \pm 0.011$ | $0.143 \pm 0.046$ | $0.278 \pm 0.064$ |
| DINOv2 (LVD-142M) | $0.183 \pm 0.008$ | $0.294 \pm 0.013$ | $0.138 \pm 0.007$ | $0.269 \pm 0.012$ | – | – |
| CLIP (WIT-400M) | $0.182 \pm 0.008$ | $0.292 \pm 0.012$ | $0.131 \pm 0.006$ | $0.268 \pm 0.012$ | – | – |
| VideoMAE (Kinetics-400) | $0.189 \pm 0.010$ | $0.318 \pm 0.013$ | $0.133 \pm 0.007$ | $0.298 \pm 0.013$ | – | – |
| VIT-M (IN) | $0.192 \pm 0.009$ | $0.321 \pm 0.013$ | $0.129 \pm 0.007$ | $0.301 \pm 0.012$ | $0.254 \pm 0.061$ | $0.446 \pm 0.100$ |
| VIT-M (IN+PT) | $0.172 \pm 0.008$ | $0.331 \pm 0.013$ | $0.148 \pm 0.007$ | $0.311 \pm 0.013$ | – | – |
| VIT-M (IN+FT) | $0.233 \pm 0.009$ | $0.345 \pm 0.014$ | $0.211 \pm 0.010$ | $0.330 \pm 0.011$ | $0.260 \pm 0.051$ | $0.461 \pm 0.099$ |
| VIT-M (IN+PT+FT) | $\mathbf{0.285 \pm 0.011}$ | $\mathbf{0.347 \pm 0.014}$ | $0.235 \pm 0.010$ | $0.328 \pm 0.013$ | – | – |
| VIT-C (IN+PT) | $0.137 \pm 0.008$ | $0.314 \pm 0.013$ | $0.120 \pm 0.006$ | $0.283 \pm 0.011$ | – | – |
| BEAST (IN+PT) | $0.205 \pm 0.008$ | $0.292 \pm 0.012$ | $0.133 \pm 0.006$ | $0.309 \pm 0.013$ | – | – |
| BEAST (IN+FT) | $0.264 \pm 0.010$ | $\mathbf{0.347 \pm 0.014}$ | $\mathbf{0.240 \pm 0.010}$ | $\mathbf{0.331 \pm 0.013}$ | $\mathbf{0.281 \pm 0.054}$ | $\mathbf{0.464 \pm 0.089}$ |
| BEAST (IN+PT+FT) | $0.282 \pm 0.011$ | $\mathbf{0.347 \pm 0.014}$ | $0.234 \pm 0.010$ | $0.326 \pm 0.013$ | – | – |

# D POSE ESTIMATION

## D.1 MODELS

The pose estimation models consist of a backbone and a head. The backbone is either a ResNet-50 (He et al., 2016) or a VIT-B/16 (Dosovitskiy et al., 2020), both producing feature maps of shape $[N, H, W]$ for a given image, where $N$ denotes the feature dimension and $H, W$ denote the height and width of the feature maps. All models employ an identical linear upsampling head that begins with a `PixelShuffle` layer, reshaping the feature maps to $[N/4, 2H, 2W]$. These reshaped features then pass through two consecutive 2D convolutional transpose layers with kernel size $(3, 3)$ and stride $(2, 2)$, doubling the spatial resolution after each layer. The head architecture omits batch normalization and nonlinearities between these layers. The output passes through a 2D softmax function, generating a normalized heatmap for each keypoint.

## D.2 TRAINING

We divided the labeled data into training (95%) and validation (5%) sets, with test frames coming from entirely held-out videos. We used a batch size of eight frames. Data augmentations include random crops, rotations, motion blur and histogram equalization. Models were trained for 300 epochs, with validation loss recorded every five epochs. For final evaluation, we selected the model with the lowest validation loss. Training utilized an Adam optimizer (Kingma and Ba, 2014) with an initial learning rate of 0.001 for the ResNet and $5e{-}5$ for the transformers, which was halved at epochs 150, 200, and 250. To facilitate feature learning, we kept the backbone frozen during the first 20 epochs of training before allowing for full end-to-end optimization. The loss function is the mean square error between each predicted heatmap and a ground truth heatmap constructed from labeled data.

## D.3 PRETRAINED BACKBONES

We test several additional VIT backbones on pose estimation to validate our BEAST results:

- Segment Anything (SAM) (Kirillov et al., 2023); checkpoint from `https://huggingface.co/facebook/sam-vit-base`

- DINO (Caron et al., 2020); checkpoint from `https://huggingface.co/facebook/dino-vitb16`

- DINOv2 (Oquab et al., 2023); checkpoint from `https://huggingface.co/facebook/dinov2-base`

These backbones are trained using the same procedure as the ResNet-50 and BEAST models (see above). We find other pretrained backbones mostly outperform the ResNet-50 baseline (Fig. 9). SAM is generally the least performant backbone. DINOv2 consistently outperforms DINO across all datasets, and BEAST achieves the lowest pixel error in most cases (only outperformed by DINOv2 in the Mirror-fish dataset). These results demonstrate BEAST's experiment-specific pretraining framework can surpass state-of-the-art general purpose vision foundation models for pose estimation.

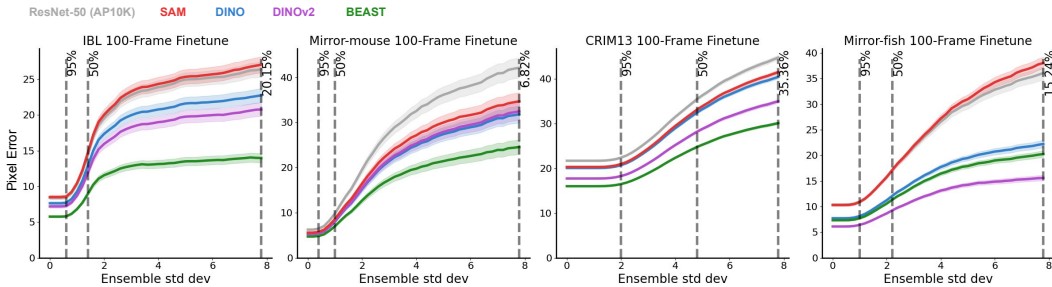

Figure 9: **Pose estimation of fine-tuned vision foundation model backbones.** We evaluated ResNet-50 (pretrained on AP10K), SAM, DINO, DINOv2, and BEAST backbones on pose estimation datasets. We evaluate the these models using pixel error at various ensemble standard deviation thresholds, with values in the table representing the percentage of keypoints at the chosen threshold. Smaller values indicate smaller but more challenging subsets of keypoints (see main text).

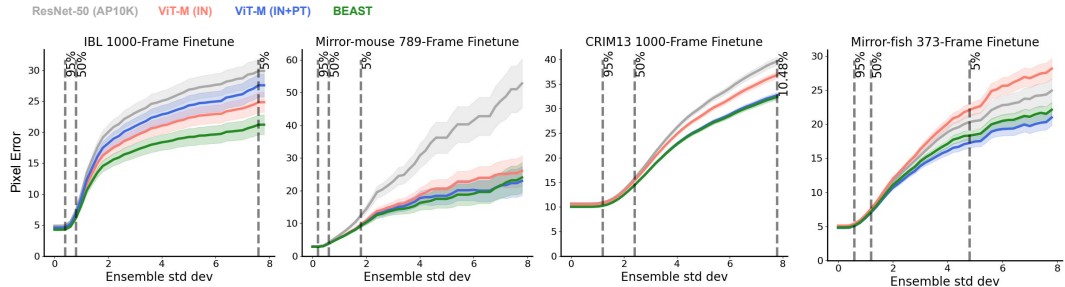

Figure 10: **Pose estimation performance with more training frames.** The results in Fig. 3 demonstrate pose estimation performance of various models using just 100 labeled frames. To ensure those comparisons hold on larger datasets we trained models (three random seeds per backbone) using the maximum number of frames in the training dataset or 1000 frames, whichever is smaller. We still see consistent gains for both the transformer architecture pretrained on ImageNet (red) and our pretraining strategy (blue, green).

### D.4 DEEPLABCUT BASELINE

For the DeepLabCut baseline (version 3.0.0) we trained models using an ImageNet-pretrained ResNet-50 backbone. To properly isolate differences between DeepLabCut and Lightning Pose algorithms, we matched training frames, batch size, learning rate schedule, and number of epochs. For all other hyperparameters we used the DeepLabCut package defaults (e.g., data augmentation). We train models using three different train/val data splits, and ensure these splits exactly match those used for the Lightning Pose models.

Lightning Pose outperforms DeepLabCut on the IBL, Mirror-mouse, and Mirror-fish datasets, while DeepLabCut outperforms Lightning Pose on CRIM13 (Fig. 11). Notably, our pretrained BEAST backbones outperform both DeepLabCut and Lightning Pose across all four datasets.

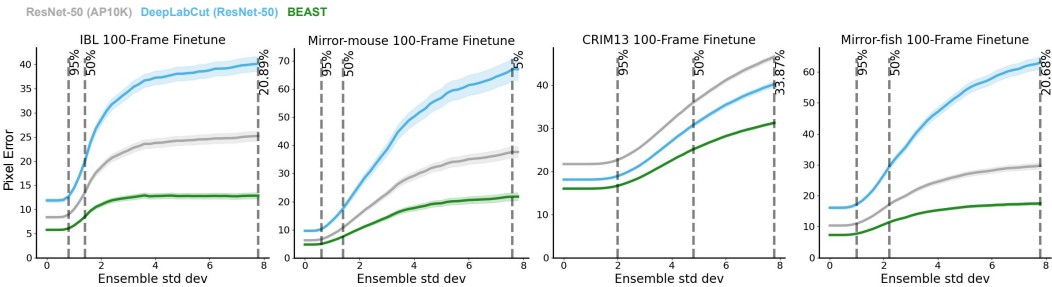

Figure 11: Pose estimation of DeepLabCut (DLC). Figure conventions as in Fig. 9

# E  ACTION SEGMENTATION

For action segmentation we consider a variety of input feature types and modeling approaches. We first consider a range input features: keypoints, PCA on the raw video frames (fit on the same frames as ViT pretraining; we use 768 PCs for downstream models), and VIT-based `CLS` tokens. For each of these feature types we fit both linear and nonlinear (temporal convolution network, TCN) models. We also train a TCN model on the VIT patch embeddings, with a multi-head attention pooling layer to reduce the dimensionality of the features before entering the TCN.

## E.1  MODELS

**Linear model.** The linear action segmentation model uses a 1D temporal convolution layer, followed by a linear layer that maps from the number of features the the number of action classes, followed by a softmax. There are no other forms of nonlinearity in the model.

**Temporal convolution network.** The nonlinear action segmentation model is a dilated TCN (Lea et al., 2016) with 2 dilation blocks. Each dilation block consists of a sequence of 2 sub-blocks (1D convolution layer → leaky ReLU nonlinearity → dropout with probability=0.10), as well as a residual connection between the input and output of the dilation block. The dilation of the convolutional filters starts with 1 for the first dilation block, then increases by a factor of 2 for each additional dilation block. This results in a larger temporal receptive field as the model gets deeper, allowing for learning of longer range dependencies (Yu and Koltun, 2015).

Both models utilize a weighted cross entropy loss function, with class weights inversely proportional to the class frequency in the training data.

## E.2  TRAINING

Each video is split into sequences of 500 (IBL) or 100 (CalMS21) frames. We divide the data into training (90%) and validation (10%) sets, with test frames coming from entirely held-out videos. We use a batch size of 16 sequences. Models were trained for 500 epochs using the Adam optimizer (Kingma and Ba, 2014).

## E.3  HYPERPARAMETER DETAILS

For each model type–linear and nonlinear–and each feature type, we run a hyperparameter search across all combinations of parameters in Table 14 using three random weight initializations. The hyperparameter combination with the best F1 score on the validation data, averaged across the three seeds, is selected for evaluation on the test set. For this hyperparameter combination, we train with two additional seeds and report results in the figures and tables averaged across five seeds.

## E.4  TEMPORAL CONVOLUTION NETWORK WITH MULTI-HEAD ATTENTION POOLING

We aggregate the patch embeddings from the BEAST encoder using a multi-head attention pooling layer (Lee et al., 2019), which produces a single pooled embedding per frame as input to the TCN.

Table 14: Action segmentation hyperparameters. *TCN only

| Hyperparameter | Value Range |
|---|---|
| Learning rate | 1e-3, 1e-4, 1e-5 |
| Dropout | 0.1 |
| Temporal filter length | 9, 17, 33 |
| Number of hidden units* | 16, 32, 64, 96 |
| Number of hidden layers* | 2 |

This layer uses a learnable query $S \in \mathbb{R}^{1 \times D}$ with patch embeddings $z \in \mathbb{R}^{N \times D}$ as keys and values. To capture motion-related features, we further concatenate the frame-to-frame difference of the pooled embeddings as additional input to the TCN (Fig. 12). We fixed the number of attention heads in the pooling layer to 8 for all models.

Our previous model utilized `CLS` embeddings, which allowed for an efficient workflow: we processed videos through the transformer backbone and saved the `CLS` embedding from each frame as a separate file. These pre-computed embeddings could then be directly loaded to train the downstream TCN classifier without requiring video reading during training. However, patch embeddings present significantly larger memory requirements, making disk storage infeasible. To address this challenge, we developed a data loading pipeline that performs end-to-end processing: it loads video frames, passes them through the transformer backbone, and feeds the resulting patch embeddings directly to the TCN model within the same training loop. This integrated approach, while computationally more intensive, eliminates the need for intermediate storage. Due to these increased computational demands, we modified the training procedure for the multi-head attention pooling models as follows.

The pooling layer and TCN classifier were trained jointly for 200 epochs on CalMS21 and 100 epochs on IBL using the Adam optimizer (Kingma and Ba, 2014) with an initial learning rate of $1e-3$. The epoch counts were determined through a separate experiment that withheld a subset of validation videos and monitored the validation F1 score until convergence. Training followed a cosine-annealing schedule with warm restarts (Loshchilov and Hutter, 2016) configured with $T_0 = 34$, $T_{mult} = 2$ and $\eta_{min} = 5e-5$. We used 6 or 8 NVIDIA A40 GPUs for the training, each with a batch size of 2, giving an effective batch size of 12 or 16. The sequence length was fixed at 500. Due to the increased compute required to train these models, we fixed the TCN hyperparameters to be those found for the `CLS`-based model for each dataset.

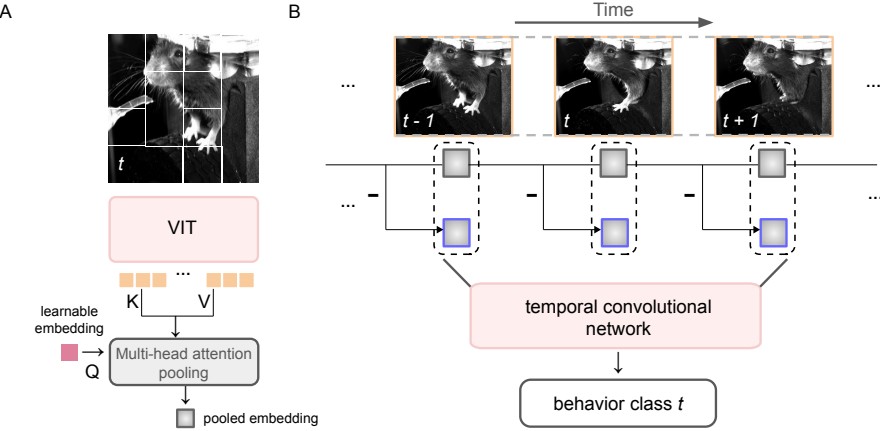

Figure 12: **Multi-head attention pooling TCN head for action segmentation. A:** Per-patch embeddings from the BEAST encoder are pooled using a multi-head attention layer, where a learnable query token attends to the patch embeddings to produce a single pooled embedding for each frame. **B:** Temporal differences between consecutive pooled embeddings are concatenated as additional input to the TCN, which predicts the behavior class of the center frame in the sliding window. A window size of 3 is shown for illustration; the actual window size was tuned as a hyperparameter.

Table 15: Action segmentation results on IBL and CalMS21 datasets across feature types and models. All VɪT-based models use frozen a frozen backbone; "CLS" indicates models trained on the global CLS embeddings, while "patch" indicates models trained with a multi-head attention pooling layer applied to the patch embeddings. "IN" refers to a model pretrained with ImageNet weights, "IN+PT" refers to models that are initialized with ImageNet-pretrained weights then further pretrained on experiment-specific data. We report the mean and standard deviation of F1 on test data across five random train/val splits.

| Dataset | Features | Linear | | TCN | |
|---|---|---|---|---|---|
| | | Features | Features, Δ Features | Features | Features, Δ Features |
| IBL | Keypoints | $0.54 \pm 1.4e{-}3$ | $0.55 \pm 1.5e{-}3$ | $0.86 \pm 1.4e{-}3$ | $\mathbf{0.88 \pm 2.2}e{-}\mathbf{3}$ |
| | PCA | $0.54 \pm 4.6e{-}3$ | $0.55 \pm 6.3e{-}3$ | $0.64 \pm 1.0e{-}2$ | $0.71 \pm 2.8e{-}3$ |
| | VIT-M (IN) (CLS) | $0.68 \pm 1.4e{-}3$ | $0.68 \pm 7.0e{-}4$ | $0.78 \pm 7.4e{-}3$ | $0.79 \pm 2.7e{-}3$ |
| | VIT-M (IN+PT) (CLS) | $\mathbf{0.74 \pm 2.5}e{-}\mathbf{3}$ | $\mathbf{0.72 \pm 2.0}e{-}\mathbf{3}$ | $0.78 \pm 4.0e{-}3$ | $0.80 \pm 4.6e{-}3$ |
| | Bᴇᴀsᴛ (IN+PT) (CLS) | $0.70 \pm 7.9e{-}3$ | $0.69 \pm 2.5e{-}3$ | $\mathbf{0.80 \pm 3.0}e{-}\mathbf{3}$ | $0.81 \pm 6.9e{-}4$ |
| | DINOv2 (patch) | - | - | - | $0.77 \pm 2.7e{-}3$ |
| | VIT-M (IN) (patch) | - | - | - | $0.84 \pm 3.7e{-}3$ |
| | VIT-M (IN+PT) (patch) | - | - | - | $0.85 \pm 3.6e{-}3$ |
| | VɪT-C (IN+PT) (patch) | - | - | - | $0.79 \pm 4.6e{-}3$ |
| | Bᴇᴀsᴛ (IN+PT) (patch) | - | - | - | $0.87 \pm 5.1e{-}3$ |
| CalMS21 | SimBA (Goodwin et al., 2024) | $\mathbf{0.75 \pm 5.1}e{-}\mathbf{4}$ | $0.53 \pm 8.1e{-}3$ | $\mathbf{0.78 \pm 3.8}e{-}\mathbf{3}$ | $0.79 \pm 2.9e{-}3$ |
| | TREBA (Sun et al., 2021b) | $0.29 \pm 1.4e{-}3$ | $0.30 \pm 1.5e{-}3$ | $0.70 \pm 4.6e{-}3$ | $0.72 \pm 7.4e{-}3$ |
| | PCA | $0.10 \pm 3.1e{-}3$ | $0.10 \pm 3.2e{-}3$ | $0.16 \pm 5.1e{-}3$ | $0.18 \pm 4.5e{-}3$ |
| | VIT-M (IN) (CLS) | $0.50 \pm 2.2e{-}2$ | $0.52 \pm 1.3e{-}3$ | $0.53 \pm 1.2e{-}2$ | $0.60 \pm 2.8e{-}3$ |
| | VIT-M (IN+PT) (CLS) | $0.60 \pm 5.5e{-}3$ | $\mathbf{0.60 \pm 1.1}e{-}\mathbf{3}$ | $0.60 \pm 9.1e{-}3$ | $0.65 \pm 2.2e{-}3$ |
| | Bᴇᴀsᴛ (IN+PT) (CLS) | $0.53 \pm 4.6e{-}3$ | $0.51 \pm 4.0e{-}2$ | $0.58 \pm 5.2e{-}3$ | $0.63 \pm 2.7e{-}3$ |
| | DINOv2 (patch) | - | - | - | $0.68 \pm 4.4e{-}3$ |
| | VIT-M (IN) (patch) | - | - | - | $0.74 \pm 2.9e{-}3$ |
| | VIT-M (IN+PT) (patch) | - | - | - | $\mathbf{0.82 \pm 9.5}e{-}\mathbf{3}$ |
| | VɪT-C (IN+PT) (patch) | - | - | - | $0.53 \pm 2.9e{-}2$ |
| | Bᴇᴀsᴛ (IN+PT) (patch) | - | - | - | $0.81 \pm 7.7e{-}3$ |

## F  Sᴇʟꜰ-sᴜᴘᴇʀᴠɪsᴇᴅ ʟᴇᴀʀɴɪɴɢ ᴍᴇᴛʜᴏᴅs ꜰᴏʀ ᴀɴɪᴍᴀʟ ʙᴇʜᴀᴠɪᴏʀ

Self-supervised learning (SSL) techniques from computer vision have increasingly been adapted to the study of animal behavior. These approaches use SSL to extract useful feature representations from pose, image, or video data, which are then applied to downstream tasks such as action segmentation.

**Pose-based approaches.** Trajectory Embedding for Behavior Analysis (TREBA) (Sun et al., 2021b) employs a multi-task self-supervised framework that uses trajectory reconstruction as its primary objective through Trajectory Variational Autoencoders (Co-Reyes et al., 2018; Zhan et al., 2020). TREBA additionally requires the TVAE embedding to decode various auxiliary tasks consisting of simple data transformations designed by domain experts, with the resulting embeddings serving as input to downstream action segmentation models. Variational Animal Motion Embedding (VAME) (Luxem et al., 2022) uses a sequential variational autoencoder to embed pose sequences into a latent space by reconstructing both current and subsequent time steps. Unlike TREBA, VAME applies clustering to the learned embeddings, creating a fully unsupervised action segmentation pipeline. ContrastivePose (Zhou et al., 2022) leverages geometric augmentations (flipping, rotation, translation) of pose coordinates to generate positive pairs for contrastive learning, followed by fine-tuning on action segmentation tasks. Bootstrap Across Multiple Scales (BAMS) (Azabou et al., 2023) employs dual temporal convolutional networks with different receptive field sizes to create complementary short- and long-term embedding spaces. BAMS introduces a novel training objective requiring prediction of future action distributions rather than specific action sequences, with validation on the MABe benchmark (Sun et al., 2023) across multiple tasks including action segmentation and mouse strain classification.

**Image and video-based approaches.** Selfee (Jia et al., 2022) constructs composite RGB frames from 3-frame grayscale video sequences (assigning each frame to a separate color channel) and applies standard image-based contrastive learning techniques, demonstrating effectiveness on action segmentation and anomaly detection. Mueller et al. (2025) adapt a pretrained V-JEPA model (Bardes et al.,

2023), an SSL approach specialized for video understanding, through domain-adaptive pretraining on primate behavior datasets, validating their approach on behavior recognition tasks. Similarly, Animal-JEPA (Zheng et al., 2024) modifies the V-JEPA training strategy with domain-specific masking techniques and validates on mouse behavior classification tasks.

**Distinction from prior work.** While these methods share conceptual similarities with BEAST, our approach is distinguished by its general frame-based training objectives and comprehensive evaluation across neural activity prediction and pose estimation tasks, in addition to the standard action segmentation task. Furthermore, since the pose-based methods described above rely on pose estimates as input, BEAST could potentially enhance their performance by providing higher-quality pose estimation as a preprocessing step.

## G BEAST WORKFLOW

We give an overview of the BEAST workflow for a new user, highlighting steps where BEAST enhances a traditional workflow.

### STAGE 1: DATA COLLECTION

- Collect behavioral videos
- Optionally collect simultaneous neural recordings

The use of BEAST does not affect this step of the workflow.

### STAGE 2: BEAST PRETRAINING

This step will be skipped in a traditional workflow.

- Extract frames from unlabeled videos (typically ∼100K frames)
- Train BEAST transformer backbone on these frames (∼12 hours on 8 GPUs)
- This step does not require manual annotation

### STAGE 3: DOWNSTREAM APPLICATIONS

*Option A: Pose estimation*

- Annotate 100-1000 frames from 5-50 videos with keypoints (can be different from pretraining set)
- Fine-tune pose estimation model using BEAST backbone
  Key advantage: improved performance over existing backbones (Fig. 3)
- Run inference on new videos to extract keypoints

*Option B: Action segmentation*

- Annotate 1000-5000 frames per behavior, ideally across 5-10 videos, with action labels (independent of pose annotation)
- Fine-tune action segmentation model using BEAST backbone
  Key advantage 1: skip pose estimation pipeline entirely
  Key advantage 2: equivalent or better performance compared to pose estimates (Fig. 4)
- Run inference on new videos to extract frame-by-frame actions

*Option C: Neural encoding*

- No manual annotation (neural activity provides the "labels")
- Fine-tune neural encoding model using BEAST backbone (video input, neural activity output)
  Key advantage 1: skip pose estimation pipeline entirely
  Key advantage 2: improved performance over existing behavioral features (Fig. 2)
- Predict neural activity from video

## H  BROADER IMPACTS

The BEAST framework enables more efficient extraction of meaningful information from video data, potentially accelerating behavioral neuroscience research with several beneficial outcomes. By reducing the need for extensive human labeling while improving accuracy, BEAST can democratize advanced video analysis capabilities for laboratories with limited resources. This efficiency could accelerate basic science discoveries that underlie advances in biomedical applications, neurological disorder treatments, and improved understanding of brain function.

While BEAST is developed primarily for behavioral neuroscience studies using animal subjects, the underlying technology could potentially be repurposed for human video analysis, raising several concerns:

- Surveillance capabilities: The improved ability to track and categorize behaviors could enhance surveillance technologies, potentially infringing on privacy rights if deployed without appropriate oversight.

- Bias and fairness: As with any AI system trained on specific datasets, BEAST-derived models may perform differently across demographic groups if applied to human subjects, potentially perpetuating biases in downstream applications.

- Resource inequality: While a pretrained BEAST model can improve the efficiency of downstream tasks, the computational requirements for pretraining itself may limit access to this technology for under-resourced institutions, potentially widening existing disparities in research capabilities.

