# OpenReview forum: "Animal behavioral analysis and neural encoding with transformer-based self-supervised pretraining"
_ICLR.cc/2026/Conference — ICLR 2026 Poster_

### Official Review · Reviewer_2yb7 · 2025-10-15

**Soundness:** 3
**Presentation:** 3
**Contribution:** 2
**Rating:** 6
**Confidence:** 4

**Summary:**

The paper suggests to use InfoNCE contrastive loss to learn better representations from the video, where similar and different embeddings are defined based on the time proximity.
The model in pre-trained jointly with InfoNCE and reconstruction (demasking)  loss (MSE) and later fine-tuned for a specific task, such as a)neural encoding; b) pose estimation; c)action segmentation.

**Strengths:**

1. The paper is very well-written and nice to read
2. The paper brings recent advances in self-supervised learning toward animal tracking community and tests the proposed method on various datasets from multiple labs.
3. On some tasks like Action segmentation (table 3) the method shows a clear improvement
4. The authors consistently provides error bars (SEM or std) in all plots and tables
5. Realistic evaluation: The limited labeling settings (100 labeled frames as described in page 7 lines 362-363) is close to real life evaluation scenarios.

**Weaknesses:**

1. A lot of references are missing, I would highly inspire the authors to do another iteration on the related works
The paper basically introduced contrastive learning on spatiotemporal video chunck, which have already been done in [1] also using InfoNCE. Additionally, Triplet loss [2] another conceptually close (to attract and repel to the anchor point) and is worth mentioning and maybe benchmarking.
On the animal tracking / pose estimation side, SLEAP [3], DeepPoseKit [4], TRex [5] are missing.
Corresponding baselines (or justification if they are not needed) is missing as well.

2. Its not clear why contrastive temporal loss even works for highly repetitive behaviour - e.g. for the dataset of a mouse walking paws - this is supposed to be very close to each other no matter is the points are close in time or not, as far as the paws are arranges same ("left or right paw step"). A bit of this discussion together with the frame selection process is is presented in the Appendix B2 but I do not find it clear enough and Table 4 and 5 often do not show statistically significant improvement (eg the improvement is 3rd digit after the dot, while the std is variable in the 1st digit after the dot, as in Table 4 1st column).

3. The improvements in tables seems to be borderline statistically significant (within std error bars, e.g. table 1 reports the improvement in the 2nd digit behind the dot while $\pm \text{std} \approx 0.1$, same comments for tables 4 and 5 was already done above).


References:
[1] Qian, Rui, et al. "Spatiotemporal contrastive video representation learning." Proceedings of the IEEE/CVF conference on computer vision and pattern recognition. 2021.
[2] Schroff, Florian, Dmitry Kalenichenko, and James Philbin. "Facenet: A unified embedding for face recognition and clustering." Proceedings of the IEEE conference on computer vision and pattern recognition. 2015.
[3] Pereira, Talmo D., et al. "SLEAP: A deep learning system for multi-animal pose tracking." Nature methods 19.4 (2022): 486-495.
[4] Graving, J. M. et al. DeepPoseKit, a software toolkit for fast and robust animal pose estimation using deep learning. eLife 8, e47994 (2019).
[5] Walter, T. & Couzin, I. D. TRex, a fast multi-animal tracking system with markerless identification, and 2D estimation of posture and visual fields. eLife 10, e64000 (2021).

**Questions:**

1. *(most crucial)* Could you provide some qualitative experiments showing that the temporal contrastive loss actually selects different enough negative pairs? Have you explored any alternative sampling strategies for the frames ?
2. In lines 482-484 you write *"The black-box nature of VIT embeddings presents interpretability challenges in scientific contexts where transparent representations like pose estimates are often preferred."*. Could you please clarify how exactly is your model more interpretable?
3. How do you match frequencies between behaviour and neuronal recordings for Fig2? Spiking neuronal data should be much higher frequency than behavioural timeseries and calcium data should be slower, especially for Facemap dataset, where behavioural videos should have been 50Hz to catch whiskers motions.
4. What is the intuition/ interpretation behind Bits Per Spike metric, especially in the calcium imaging settings?
5. Why SLEAP and DeepLabCut are not used as baselines for pose estimations? (DeepLabCut might be seen as pretrained ResNet version but having simple UNet comparison is missing, which both of them do, is missing). If you have tried it and it did not work on 100 frames, I think it should be mentioned in the appendix given how common these tools are in the community.
6. How would you expect you method to work for animal tracking for in the wild (e.g. monkeys tracking in the jungles), where the motion is much less periodic and more various but the background could be stationary as well (e.g. wind would move the leaves)?

Minor:
1. You report training *"taking approximately 25 hours on 8 Nvidia A40 GPUs"* in line 202. This is without ImageNet pretraining and simultaneously on all datasets (e.g. mice and fish together)? Why does it take so long if you start from the weights after ImageNet pretraining? Do you have a warmup stage before cosine annealing?
2. Why do you train CEBRA models not jointly across sessions ? (as mentioned in App. C)
3. For Fig.4 - could you show non-normalized confusion matrixes (panel B) because it looks like for the middle just the majority class gets switched? Could you also please provide the weighted accuracy ?
4. For Fig.4 panel A - what is ensembled for PCA? should not it be deterministic if only seeds are varied? If yes - how is there a difference for "CalMS21" dataset?

---

> ### Author Response · Authors · 2025-11-21
>
> We thank the reviewer for recognizing the strengths of our work, and try to clarify any remaining concerns with our comments below.
>
> **W1: Missing references.**
> We thank the reviewer for pointing out missing references. We will update the “Self-supervised learning for images and videos” section of the Related Works to include the references 1-2, along with the references shared by reviewer 6xX2. We will update the “Large-scale models for behavioral video analysis” section of the Related Works to include references 3-5. Furthermore, we note that we will include a new Related Works section dedicated to self-supervised learning for animal behavior analysis, as suggested by reviewer ys7M.
>
> **W2/Q1: Temporal contrastive loss.**
> The reviewer raises an important point about how contrastive loss functions in highly repetitive behavior. We note that even in datasets with repetitive task structure (e.g., IBL, mirror-mouse), animals exhibit substantial behavioral variability beyond the primary repetitive actions. For example, in the IBL dataset, mice engaged in the decision-making task also exhibit non-task behaviors: they groom between trials, shift posture over tens of trials (Findling et al., 2025, Extended Data Fig. 9c), and fidget in ways unrelated to the task (Musall et al., 2019). Similarly, in the mirror-mouse dataset, mice not only run on the treadmill but periodically jump to avoid obstacles. Obstacle height varies across trials, eliciting different avoidance strategies and therefore distinct behavioral patterns. This natural variability provides sufficiently diverse frames for the contrastive loss to learn meaningful temporal structure.
> To address the reviewer's concern more directly, we are developing visualizations to demonstrate the diversity of frames selected by our current sampling strategy. These will show that even in "repetitive" datasets, the contrastive loss identifies meaningfully different negative pairs.
>
> Regarding alternative sampling strategies, we explored a random frame selection approach. We acknowledge that Table 5 shows minimal quantitative differences between strategies. We are incorporating the random strategy into our visualizations to provide qualitative comparisons that may reveal more subtle differences not captured by the downstream task metrics alone.
>
> **Q2: Interpretability.**
> We apologize for the confusion - we intended to acknowledge that our model is _not_ interpretable and that future work should prioritize interpretability for scientific understanding. We will clarify this point in the updated manuscript.
>
> **Q3: Behavior/neural syncing.**
> For RRR neural encoding models, we use different implementations depending on the dataset.
> For the IBL/IBL-whisker RRR models, we use an implementation that has previously been used with IBL data: https://github.com/yzhang511/neural_decoding. This model maps the entire behavioral trial $X \in \mathbb{R}^{B \times T_b}$ (where $B$ is the behavioral representation dimensionality and $T_b$ is the number of behavioral timepoints) directly to the entire neural trial $Y \in \mathbb{R}^{N \times T_n}$ (where $N$ is the number of neurons and $T_n$ is the number of neural timepoints). Different sampling rates for behavior and neural activity result in $T_b \neq T_n$. Crucially, this model does not map individual behavioral timepoints to individual neural timepoints, nor does it use temporal windows; rather, it learns a mapping from the full behavioral trial to the full neural trial.
>
> For the Facemap RRR model, we use the implementation from the Facemap repository: https://github.com/MouseLand/facemap/blob/main/facemap/neural_prediction/prediction_utils.py#L110. This model first resamples the behavioral timestamps to match the neural timestamps, and then fits a Reduced Rank Regression model using low-rank SVD.
>
> For TCN neural encoding models, we use the implementation from the Facemap repository: https://github.com/MouseLand/facemap/blob/main/facemap/neural_prediction/neural_model.py. This model applies temporal convolutions at the full behavioral frame rate, then samples the resulting latent representation at neural timepoints using nearest-neighbor indexing (i.e., the latent from the nearest behavioral frame is used for each neural timepoint). In the IBL/IBL-whisker datasets, video is acquired at 60 Hz and binned to 50 Hz (20 ms bins), while neural activity is binned at the same 50 Hz rate. In the Facemap dataset, video is acquired at 50 Hz while neural activity is sampled at 3 Hz.
>
> We will expand Appendix C to include these details and refer readers to the Methods sections in the corresponding papers for additional implementation information.

---

> > ### Author Response · Authors · 2025-11-21
> >
> > **Q4: Neural encoding metrics.**
> > Bits per spike (BPS) measures the mutual information between predicted and observed spike counts, normalized by the total number of spikes. While other metrics like pseudo-R² could be used for spike count data, we chose BPS because it has been adopted as the standard metric by the Neural Latents Benchmark (Pei et al., 2021). Intuitively, a positive BPS indicates that the model captures a neuron's time-varying activity better than a constant mean firing rate baseline (which yields BPS=0), making it conceptually similar to R² in spirit. We note that we only use BPS for the spiking datasets IBL and IBL-whisker; for the calcium imaging dataset Facemap we use Variance Explained (see Fig. 2).
> >
> > **Q6: BEAST in the wild.**
> > Applying BEAST to videos from naturalistic environments (e.g., home cages, zoos, or the wild) represents an exciting direction for future work. We believe the core pretraining method (masked autoencoding plus temporal contrastive loss applied to a vision transformer) is well-suited to this extension. The primary challenge, as the reviewer notes, will be adapting to different visual statistics and behavioral distributions.
> >
> > We expect the frame selection strategy to require the most scrutiny. Our current approach uses PCA and k-means to identify diverse frames, which works well for laboratory videos with static backgrounds where diversity primarily reflects the animal's posture. In dynamic environments, however, this strategy would also capture background changes (e.g., lighting shifts, camera motion, moving objects), which may or may not be desirable depending on the downstream task.
> > If the goal is to focus on postural diversity while ignoring background variation, alternative preprocessing could be valuable. For example, methods like DUSt3R (Wang et al., 2023) or VGGT (Wang et al., 2025) could first extract rough 3D point clouds of the animal, enabling postural clustering on the animal representation rather than raw pixels. We will incorporate some of these observations into an updated Discussion.
> >
> > **M2: CEBRA training.**
> > We trained single-session CEBRA models to align with the other methods in Fig. 2: PCA is computed per session, and BEAST is fine-tuned per session. However, training a joint CEBRA model across all sessions represents another reasonable baseline, and a similar question was raised by reviewer 6xX2.
> >
> > To address this, we trained a multi-session CEBRA model using ViT CLS tokens as input, which also accounts for the stronger feature extraction of the ViT backbone. As described in our response to reviewer 6xX2: we find that using pretrained visual tokens does not improve standard CEBRA performance on the IBL neural encoding task. For IBL-whisker, ViT-CEBRA shows modest improvement over baseline CEBRA but remains substantially below BEAST's performance.
> >
> > **M4: Ensembled PCA.**
> > We apologize for the confusion; ensembling occurs at the action segmentation model level, not at the representation level. For each representation method (keypoints, PCA, BEAST, etc.), we train a single model to extract features. We then train multiple action segmentation models (TCNs) on these fixed representations, each with different random weight initializations, and ensemble their predictions.
> >
> > The observation that the average F1 across ensemble members equals the ensembled model F1 for the IBL dataset is coincidental. We will clarify the ensembling procedure in the final manuscript.
> >
> > **Work in progress.**
> > We are still working on responding to the following comments:
> > * W2/Q1: Temporal contrastive loss
> > * W3: Statistical significance
> > * Q5: Pose estimation baselines/DLC
> > * M1: Training time
> > * M3: Non-normalized confusion matrices

---

> > > ### Comment · Reviewer_2yb7 · 2025-11-21
> > > **Question about specific changes**
> > >
> > > Thanks a lot for your reply!
> > >
> > > Short question - do I get it correctly that the pdf is not updated yet and I cannot see the specific changes in the *related works* section and the visualisations mentioned for *W2/Q1*.
> > >
> > > *M2:* Have you tuned CEBRA? if yes, what specific tuning procedure / hyperparameter search have you done?

---

> > > > ### Author Response · Authors · 2025-11-21
> > > >
> > > > Thank you for the fast response!
> > > >
> > > > You are correct that the pdf is not yet updated to reflect changes in the Related Work section and the visualizations - we will alert you when these changes have been incorporated.
> > > >
> > > > Regarding CEBRA, we did not carry out a hyperparameter search, but used the package defaults. Instead, we focused on extensive hyperparameter searching for the downstream neural encoding models (see Appendix C). We will try fine-tuning CEBRA if time allows during the rebuttal period.

---

> > > > > ### Comment · Reviewer_2yb7 · 2025-11-23
> > > > >
> > > > > In the meanwhile I have some additional questions for the reply above
> > > > >
> > > > > **Q3** Do I get correctly that the IBL model predicts single 1 sample of behaviour variables for 1 sample of responses, where a sample corresponds to the timebin / e.g. timepoint ( I assume spiking neuronal activity is binned)?
> > > > > If yes - do you just match the binning to match the behaviour sampling frequency?
> > > > >
> > > > > **Q4** while bits per spike is close in spirit to R^2, which of them 2 is a more conservative metric?
> > > > > As for variance explained, why specifically have you chosen this metric? To the best of my knowledge, neural calcium signals are very noise and if variance-based metrics are used, they usually try to account for a fraction of explainable variance (enough of metrics like this exist - see  Pospisil & Wyeth 2021 [1], section *Comparison to prior methods*). Do I get this right that you cannot account for noise due to the lack of exact repeats in the behaviour? And if yes, why not to use correlation then, which is used in the recently established benchmarks for predicting calcium activity (Sensorium 2022 / 2023, [2,3] )
> > > > >
> > > > >
> > > > > References:
> > > > > [1] Pospisil, Dean A., and Wyeth Bair. "The unbiased estimation of the fraction of variance explained by a model." PLoS computational biology 17.8 (2021): e1009212.
> > > > > [2] Willeke, Konstantin F., et al. "Retrospective on the SENSORIUM 2022 competition." NeurIPS 2022 Competition Track. PMLR, 2023.
> > > > > [3] Turishcheva, Polina, et al. "Retrospective for the Dynamic Sensorium Competition for predicting large-scale mouse primary visual cortex activity from videos." Advances in Neural Information Processing Systems 37 (2024): 118907-118929.

---

> > > > > > ### Author Response · Authors · 2025-11-25
> > > > > >
> > > > > > **Q3. IBL RRR model.**
> > > > > > The RRR model for IBL data is fit on **per-trial** data (not per-timepoint data). The neural activity Y has shape $\(K, T_n, N\)$, where $K$ is the number of trials, $T_n$​ is the number of neural timepoints per trial, and $N$ is the number of neurons. The behavior activity $X$ has shape $\(K, T_b, B\)$, where $T_b$ is the number of behavioral timepoints per trial and $B$ is the behavioral dimensionality (e.g., number of keypoints, dimensionality of ViT CLS tokens, etc.). For each trial $k$, the RRR model is given by $Y_k = X_k (UV)’$, where $U$ has shape $\(N, R\)$ and $V$ has shape $\(R, T_n, T_b, B\)$. The formulation accommodates different values of $T_n$ and $T_b$​, so the neural and behavioral data do not need to share the same time binning or sampling frequency.
> > > > > >
> > > > > > Intuitively, $V$ maps the behavior into a set of  temporal (behavioral) basis functions, which are then mapped to a set of temporal (neural) basis functions. $U$ then maps these temporal (neural) basis functions to the observed neural activity.
> > > > > >
> > > > > > However, as mentioned in our previous comment, we note that for the IBL dataset we have already resampled the behavioral data to the same frequency as the neural data before fitting the RRR model, similar to the Facemap RRR model. The reason for using two separate RRR implementations was to match previous work with each dataset. We will clarify these points in the updated manuscript.
> > > > > >
> > > > > > **Q4: BPS vs R2.**
> > > > > > BPS and pseudo-R² can be understood as different normalizations of the same underlying quantity (log-likelihood under a Poisson observation model). In this sense, neither is inherently more conservative; rather, they measure slightly different aspects of model performance.
> > > > > >
> > > > > > BPS is defined as
> > > > > > $$\text{BPS} = \frac{1}{N_{sp} \log(2)}(LL_{model} - LL_{null})$$
> > > > > > where $N_{sp}$ is the total number of spikes and $LL_{null}$ is the log-likelihood of a constant-rate null model (Pei et al., 2022).
> > > > > > Pseudo-R² is defined as:
> > > > > > $$\text{pseudo-}R^2 = 1 - \frac{LL_{model}}{LL_{null}}$$
> > > > > > BPS emphasizes per-spike efficiency (normalized by spike count), whereas pseudo-R² measures overall fractional improvement in log-likelihood.
> > > > > >
> > > > > > We emphasize that these closely related metrics should not be directly compared to one another; rather, models should be compared using one metric consistently. We do not claim BPS is the “better” or more conservative metric, but chose it to align with standard practice in neural encoding (following its adoption in the Neural Latents Benchmark; Pei et al., 2022).
> > > > > > If the reviewer is interested in a more nuanced discussion of these metrics, we recommend Williamson et al., The Equivalence of Information-Theoretic and Likelihood-Based Methods for Neural Dimensionality Reduction.
> > > > > >
> > > > > > **Q4: Variance explained vs correlation for calcium imaging data.**
> > > > > > We apologize for the lack of clarity regarding this metric. Following the original Facemap paper (Syeda et al., 2023), rather than directly predicting per-neuron activity (sessions contain 30,000 to 50,000 simultaneously recorded neurons), we reduce the dimensionality of the neural data to 128 principal components and predict these latent dimensions from behavioral features. We will update the text and Fig. 2 to more clearly state this.
> > > > > >
> > > > > > As described in the Facemap Methods section:
> > > > > > > "To predict the neural activity from behavior, we reduced the dimensionality of the z-scored activity using singular value decomposition (SVD) and keeping 128 components… If fewer than 200 neurons were predicted, then we directly predicted the neurons rather than using the PCs. When predicting more neurons, we found that predicting the neural PCs performed and/or outperformed direct neural prediction."
> > > > > >
> > > > > > This approach addresses the reviewer's concern about noisiness of calcium signals, as we report variance explained in the neural PCs rather than in raw single-neuron activity. We note that we cannot compute the fraction of explainable variance because neural activity was recorded during spontaneous behavior (i.e., there is no repeated stimulus condition from which to estimate signal variance).
> > > > > >
> > > > > > Our setup, including data preprocessing (z-scoring followed by PCA), baseline models (RRR and TCN using the Facemap GitHub repository), and evaluation metric (variance explained of neural PCs), is designed to make our results directly comparable to the original publication.

---

> > > > > > > ### Author Response · Authors · 2025-11-26
> > > > > > >
> > > > > > > All changes outside of Appendix H are highlighted in blue in the updated pdf.
> > > > > > >
> > > > > > > **W1: Missing references.**
> > > > > > > We have updated the Related Works section to include the Recasens reference: “The contrastive method has also been extended to the temporal (Hyvarinen and Morioka, 2016) and video domain (Qian et al., 2021; Recasens et al., 2021; Dave et al., 2022).”
> > > > > > >
> > > > > > > We have also updated the Related Works section to include more pose estimation references: “DeepLabCut (Mathis et al., 2018) leverages ImageNet-pretrained backbones for fine-tuning on experiment-specific labeled datasets, offering flexibility but requiring more manual labeling. This work inspired a range of other general-purpose animal pose estimation tools including LEAP (Pereira et al., 2019), DeepPoseKit (Graving et al., 2019), TRex (Walter and Couzin, 2021), SLEAP (Pereira et al., 2022), and Lightning Pose (Biderman et al., 2024).”
> > > > > > >
> > > > > > > **W3: Statistical significance.**
> > > > > > > Two reviewers brought up concerns about significance in the neural encoding results. We appreciate this attention to detail and copy the same text for response to both reviewers for ease of reading.
> > > > > > >
> > > > > > > For the results in our tables we first computed the mean BPS per session (call this mBPS), then provide the mean and the standard deviation of the mBPS over five sessions for each dataset. This relatively small sample size of sessions leads to large standard deviations. To more carefully measure the differences between models on the full neural populations with the IBL dataset, we performed a two-sided Wilcoxon signed-rank test at the level of individual neurons (N=842 pairs) for the following models:
> > > > > > > - DINOv2
> > > > > > > - ViT-M (IN): a ViT pretrained on Image-Net with MAE loss
> > > > > > > - ViT-C (IN+PT): ViT-M further pretrained on domain-specific data with the contrastive-only loss
> > > > > > > - ViT-M (IN+PT): ViT-M further pretrained on domain-specific data with the MAE loss
> > > > > > > - BEAST (IN+PT): ViT-M with additional domain-specific pretraining using MAE and contrastive losses
> > > > > > >
> > > > > > > The detailed results are in Appendix H.4: Figs. 12 and 13, and Table 17. We find that BEAST (IN+PT) generally significantly outperforms all other models, though it is not significantly different from ViT-M (IN+PT) on either dataset when using the TCN. Table 17 contains the recomputed means and standard errors over all neurons rather than session averages (i.e. what is shown in Table 14). We will continue to update Table 17 with more baselines throughout the rebuttal period.
> > > > > > >
> > > > > > > **Updated discussion.**
> > > > > > > We have updated the Discussion to address multiple reviewer comments, including the clarification about interpretability of the ViT model and BEAST in the wild.
> > > > > > >
> > > > > > > **Q3: Behavior/neural syncing.**
> > > > > > > We have updated Appendix C to more clearly state how neural/behavioral syncing is handled by the different models.
> > > > > > >
> > > > > > > **M1: Training time.**
> > > > > > > We agree with the reviewer that training time remains a bottleneck, and addressing this represents one of our most important directions for future work. We want to clarify that the 25 hours on 8 A40 GPUs is for *each individual dataset*, starting from pretrained ImageNet weights. We found that this amount of training is necessary to achieve good image reconstructions. To demonstrate this, we evaluate reconstruction quality at 0, 200, 400, and 800 epochs during pretraining on the IBL dataset (Appendix H.3, Fig. 11). Epoch 0 represents zero-shot reconstruction quality from a model pretrained with masked autoencoding on ImageNet only.
> > > > > > >
> > > > > > > We have expanded the Discussion to address this limitation and explore potential solutions:
> > > > > > > - Smaller architectures: Pretraining smaller models like ViT-Small (approximately one-quarter the parameters of our ViT-Base) would reduce training, inference, and fine-tuning time. Thoroughly exploring accuracy-efficiency tradeoffs across architectures is crucial to making these models accessible to practicing scientists.
> > > > > > > - Multi-dataset pretraining: Rather than requiring individual labs to pretrain from scratch, we aim to one day pretrain general foundation models that downstream users can simply fine-tune. This will require balancing specificity and generalizability. For example, we might begin by pretraining on constrained settings (e.g., head-fixed mouse experiments or top-down views of freely moving mice) and gradually broaden the scope of pretraining data.
> > > > > > >
> > > > > > > [We note that our cosine annealing schedule includes a warmup phase (low → high → low learning rate); full details are in Appendix B.2.]

---

> ### Author Response · Authors · 2025-11-26
>
> **Q5: Baselines for pose estimation/DeepLabCut.**
> Two reviewers requested DeepLabCut baselines, and we copy the same response to both reviewers for ease of reading.
>
> We thank the reviewers for this suggestion, as appropriate and widely used baselines are essential for demonstrating the efficacy of our approach. We trained DeepLabCut models on our pose estimation datasets using the default ResNet-50 backbone. To isolate differences in the underlying algorithms, we matched training frames, batch size, learning rate schedule, and number of epochs with our Lightning Pose models. For all other hyperparameters, we used the respective package defaults.
>
> Lightning Pose outperforms DeepLabCut on the IBL, Mirror-mouse, and Mirror-fish datasets, while DeepLabCut outperforms Lightning Pose on CRIM13. Notably, our pretrained BEAST backbones outperform both DeepLabCut and Lightning Pose across all four datasets. We provide a summary of results below, with full pixel error versus ensemble standard deviation plots in Appendix H.5 (Fig. 14) and reference to this baseline in the main text.
>
> | Model                 | IBL         | Mirror-mouse | CRIM13       | Mirror-fish  |
> |-----------------------|-------------|--------------|--------------|--------------|
> | resnet50 (DLC)        | 11.86 ± 0.4 | 9.61 ± 0.42  | 18.13 ± 0.23 | 16.14 ± 0.46 |
> | resnet50 (LP)         | 8.4 ± 0.25  | 6.25 ± 0.18  | 21.72 ± 0.2  | 10.39 ± 0.23 |
> | BEAST                 | **5.77 ± 0.15** | **4.72 ± 0.13** | **16.02 ± 0.12** | **7.34 ± 0.12**  |
>
> **Work in progress.**
> We are still working on responding to the following comments:
> - W2/Q1: Temporal contrastive loss visualizations
> - M3: Non-normalized confusion matrices

---

> > ### Author Response · Authors · 2025-12-03
> >
> > **W2/Q1: Temporal contrastive loss visualizations.**
> > To address the comment about whether or not there is sufficient diversity in behavioral neuroscience datasets with repetitive behavior, we visualize the UMAP embeddings of the first 32 principal components of all anchor frames used for pretraining on the IBL dataset (Fig. 16). During training, only the frames immediately adjacent to each anchor were selected as positive pairs, while frames from different sessions or from more distant points within the same session served as negative samples. The figure clearly shows that frames from different sessions are highly dissimilar, owing to differences in mouse appearance, experimental equipment, and lighting. Frames within a session exhibit a high level of diversity, even frames next to each other in UMAP space (e.g., frames B and C, or D and E, on the left-hand side of Fig. 16). As a result, the frames that co-occur with an anchor in the same batch are typically visually dissimilar and therefore appropriate for our contrastive loss.
> >
> > **M3: Non-normalized confusion matrices.**
> > We have now included the non-normalized confusion matrices for the BEAST models in Appendix H.8 (Fig. 17).

---

### Official Review · Reviewer_ys7M · 2025-10-24

**Soundness:** 3
**Presentation:** 4
**Contribution:** 3
**Rating:** 6
**Confidence:** 4

**Summary:**

This paper introduces BEAST, a self-supervised learning framework that combines masked autoencoding (MAE) and temporal contrastive learning to pretrain vision transformers on animal behavior videos. The authors demonstrate the utility of BEAST across three key neuro-behavioral tasks: neural encoding, pose estimation, and action segmentation. The method is evaluated on multiple datasets and species, showing consistent improvements over existing baselines, especially in settings with limited labeled data.

**Strengths:**

Comprehensive Evaluation: The paper thoroughly evaluates BEAST across multiple tasks (neural encoding, pose estimation, action segmentation) and datasets (mice, fish), demonstrating its versatility and robustness.

Neural Encoding Focus: The inclusion of neural encoding as a downstream task is a significant and compelling contribution, linking behavioral video analysis directly to neural activity—a fundamental goal in neuroscience.

Efficient Use of Unlabeled Data: BEAST effectively leverages unlabeled video data through self-supervised pretraining, reducing the need for extensive manual annotation.

Ablation Studies: The paper includes extensive ablations (e.g., frame selection, pooling strategies, loss weighting) that provide valuable insights into the design choices.

**Weaknesses:**

1. Limited Discussion of Related Self-Supervised Methods: While the paper discusses general self-supervised learning (SSL) methods like MAE and contrastive learning, it does not adequately address existing SSL approaches specifically designed for animal behavior. For example:

    ConstrastivePose: A contrastive learning approach for self-supervised feature engineering for pose estimation and behavorial   classification of interacting animals.
    Tianxun Zhou, Calvin Chee Hoe Cheah, Eunice Wei Mun Chin, Jie Chen, Hui Jia Farm, Eyleen Lay Keow Goh, Keng Hwee Chiam.
    bioRxiv 2022.11.09.515746; doi: https://doi.org/10.1101/2022.11.09.515746

    Selfee: Self-supervised Features Extraction of animal behaviors.
    Yinjun Jia, Shuaishuai Li, Xuan Guo, Junqiang Hu, Xiao-Hong Xu, Wei Zhang.
    bioRxiv 2021.12.24.474120; doi: https://doi.org/10.1101/2021.12.24.474120

    Domain-Adaptive Pretraining Improves Primate Behavior Recognition.
    Felix B. Mueller, Timo Lueddecke, Richard Vogg, Alexander S. Ecker.
    arXiv:2509.12193; doi: https://doi.org/10.48550/arXiv.2509.12193

    A more thorough discussion of these works would better contextualize BEAST’s contributions.

2. Performance on IBL-Whisker Dataset: The results show that BEAST does not outperform DINOv2 on the IBL-whisker dataset (Table 4). The authors should discuss why this might be the case (e.g., DINOv2’s stronger pretraining or architectural advantages?). Also, whether further pretraining DINOv2 on IBL data would close or widen the gap.

3. Missing DINOv2 Baselines for some tasks: While DINO is included in pose estimation and DINO v2 is evaluated in neural encoding, DINOv2 is not evaluated for behavior classification or pose estimation. It is confusing. Given its strong performance in other domains, it should be included as a baseline for a fair comparison.

4. Technical Novelty: The core components of BEAST (MAE + contrastive learning) are based on ViC-MAE, which is not novel. The main contributions lie in the application-specific adaptations (e.g., frame sampling, domain-specific pretraining) rather than the underlying architecture.

**Questions:**

1. Why does BEAST largely underperform compared to DINOv2 on the IBL-whisker dataset (in terms of RRR)? Would pretraining DINOv2 on IBL data improve its performance further?

2. Have you considered using only patches containing the animal (via segmentation or attention) for behavior classification? This may help reduce background noise.

3. Why was DINOv2 not included as a baseline for behavior classification and pose estimation? Its strong performance in other domains warrants its inclusion.

4. How does BEAST compare to DINOv3, which was released after this submission?

---

> ### Author Response · Authors · 2025-11-21
>
> We thank the reviewer for recognizing the strengths of our work, and try to clarify any remaining concerns with our comments below.
>
> **W1: Limited discussion of behavior SSL.**
> We thank the reviewer for pointing out these references, and agree that an expanded discussion of SSL methods designed for animal behavior is important to properly situate our work in the literature. We will incorporate these references into a new section in the Related Works.
>
> **W2/Q1: DINOv2 performance on IBL-whisker dataset.**
> The RRR performance of DINOv2 on the IBL-whisker dataset is indeed an outlier in Table 4. While stronger pretraining could explain this result, this explanation does not account for the IBL dataset, where BEAST strongly outperforms DINOv2 despite both models being pretrained. We suspect the highly out-of-distribution nature of IBL-whisker (Fig. 2A, second row) plays a significant role. We are currently investigating feature visualization methods (see response to reviewer gTWQ's interpretability comment) and will apply these to the IBL-whisker dataset to better understand this result.
>
> Based on our comparisons between ViT-MAE (pretrained on ImageNet) and BEAST (further pretrained on domain-specific data), we expect that further pretraining DINOv2 on IBL-whisker data would likely improve performance. However, there is an important technical caveat: our MAE-based loss requires a pixel-level decoder (~10M parameters). When starting with ViT-MAE, we leverage a pretrained decoder, but DINOv2 lacks such a decoder. Therefore, domain-specific pretraining of DINOv2 using our MAE loss would require randomly initializing a decoder, which adds complexity and may diminish the benefits of starting from a stronger backbone. This is why we have focused on further pretraining models that were initially trained with MAE loss.
>
> Nevertheless, we agree that investigating stronger backbones like DINOv2 represents an interesting direction for future work. We see our work as a first step toward exploring domain-specific adaptation of powerful self-supervised objectives, including MAE, DINO, and I-JEPA. We hope our results establish a strong foundation for MAE-based models that can be extended and compared to future computer vision advances.
>
> **W4: Technical novelty.**
> We agree with the reviewer that our main contribution lies in domain-specific adaptation of existing self-supervised objectives, and will clarify this in the text. We hope the reviewer sees value in this form of contribution, particularly in light of the clear gap in accessible, general-purpose models for scientific video analysis.
>
> **Q2: Using patches containing only the animal.**
> The suggestion to use only patches containing the animal(s) for behavior classification is an interesting one that we have not yet explored. This approach would be relatively straightforward for head-fixed datasets like IBL, where the animal occupies a consistent set of patches across frames. However, it would be more challenging for freely moving datasets like CalMS21, where animals constantly move through different patches, requiring frame-by-frame identification of animal-containing patches.
> One approach would be to use a preprocessing step: run a detection algorithm to obtain bounding boxes for each animal, translate those boxes into patches, and drop non-overlapping patches before classification.
>
> A more integrated approach would leverage our pretrained backbone for detection as a fourth downstream task. The pipeline would work as follows: (1) frames are processed through the backbone; (2) features are used to generate bounding boxes (with performance potentially improved by domain-specific pretraining); (3) bounding boxes are used to mask decoder output tokens; and (4) unmasked tokens are fed to the action segmentation head. This multi-task approach represents an interesting direction for future work.
>
> **W3/Q3: DINOv2 baselines.**
> We agree with the reviewer that DINOv2 should be included as a baseline for pose estimation and action segmentation. We have implemented this baseline for pose estimation, with results in Appendix H.2. We find that DINOv2 consistently outperforms (or matches) DINO across all datasets. However, BEAST still significantly outperforms DINOv2 on three datasets: IBL-paw, Mirror-mouse, and CRIM13. On the Mirror-fish dataset, DINOv2 outperforms BEAST (notably, this dataset already showed the smallest performance gap between DINO and BEAST).
>
> We are currently working on implementing DINOv2 for action segmentation and will provide results as soon as they are available.
>
> **Q4: DINOv3 baselines.**
> We appreciate the suggestion to compare with DINOv3. Although DINOv3 was released after our initial submission, we agree it represents an interesting and powerful baseline. If time permits during the rebuttal period, we will attempt to benchmark this model on a subset of our downstream tasks and include preliminary results.

---

> ### Author Response · Authors · 2025-11-21
>
> **Work in progress**
> We are still working on responding to the following comments:
> * W1: Limited discussion of behavior SSL
> * W3/Q3: DINOv2 baseline for action segmentation

---

> ### Comment · Reviewer_ys7M · 2025-11-22
>
> Thank you for the detailed and helpful responses to my review comments.
>
> I am generally satisfied with your clarifications and the additional results provided in the Appendix, particularly the DINOv2 baseline. I look forward to the ongoing results for a complete picture.
>
> I agree this work holds significant value for neuro-behavioral research. Therefore, I maintain that this is a solid work, but existing SSL models like MAE and general-purpose models like the DINO series may shade the novelty of this work.
>
> Finally, regarding DINOv2's lack of a decoder for MAE loss: I wish to clarify that DINOv2 is built on an iBOT-inspired architecture, which includes a masked prediction objective. While its head may not be pixel-level, it does utilize a transformer-based "iBOT head" to predict features of masked tokens, effectively serving as an MAE-like decoder for feature-level reconstruction.
>
> Overall, the paper is stronger after the rebuttal, and I maintain my marginally above acceptance rating.

---

> ### Author Response · Authors · 2025-11-26
>
> We thank the reviewer for engaging in discussion with us, and their suggestions have strengthened our work. Below are our responses to the remaining questions, but we are of course happy to address any further comments/questions/concerns.
>
> **W1: Limited discussion of behavior SSL.**
> We now provide an extensive related works section on SSL for animal behavior in Appendix H.2, highlighting the references shared by the reviewer as well as other relevant references. Depending on page constraints we will either put this text in the Related Works section of the main text, or reference an Appendix section from the Related Works section in the final manuscript. If the reviewer feels we have missed any other important references we are happy to further update the text.
>
> **W3/Q3: DINOv2 baseline for action segmentation.**
> Two reviewers were interested in DINOv2 results for action segmentation, so we copy the same text to each for ease of reading.
>
> We thank the reviewers for pointing out this gap in our baselines. We have now fine-tuned a DINOv2 model on action segmentation using patch embeddings, following the same procedure as for our other pretrained ViT backbones. We find that DINOv2 performs worse (F1 scores, higher is better) than both the ImageNet-pretrained ViT-MAE and BEAST:
>
> | Model           | IBL         | CalMS21      |
> |-----------------|-------------|--------------|
> | ViT-M (IN)      | 0.84 ± 0.00 | 0.74 ± 0.00  |
> | DINOv2          | 0.77 ± 0.00 | 0.68 ± 0.00  |
> | BEAST          | **0.87 ± 0.01** | **0.81 ± 0.01**  |
>
> We used identical hyperparameters across all three models due to computational constraints. DINOv2's performance may improve with additional hyperparameter tuning, which we will continue to explore within the remaining rebuttal period.
> With this result, we have now incorporated the DINOv2 baseline across all downstream tasks. This comprehensive comparison strengthens our claims about the benefits of domain-specific pretraining for behavioral neuroscience video data.
>
> Summary of BEAST vs. DINOv2 across tasks:
> - **Neural encoding**: BEAST slightly outperforms DINOv2 (Table 4)
> - **Pose estimation**: BEAST significantly outperforms DINOv2 on 3 of 4 datasets (Fig. 7)
> - **Action segmentation**: BEAST significantly outperforms DINOv2 on both datasets (Table 16)
>
> Qualitative feature analysis (Fig. 10) reveals that DINOv2 features are visually distinct from those of MAE-based backbones (ImageNet-pretrained ViT-MAE and BEAST). Further work will be required to more thoroughly assess how different pretraining strategies affect performance on neuroscience-specific downstream tasks.
>
> **DINOv2/iBOT patch-level objective.**
> We appreciate the reviewer's clarification about DINOv2's iBOT head, and we agree that exploring patch-level self-supervised objectives beyond pixel reconstruction represents an exciting direction for behavioral video analysis.
>
> We are actively exploring alternative SSL methods beyond MAE. For example, we have experimented with I-JEPA (Assran et al., 2023), but found that these approaches require more careful hyperparameter tuning that often does not transfer well across datasets. In contrast, the MAE loss in BEAST has proven robust to hyperparameter choices and dataset variations, making it easier to apply broadly without extensive per-dataset optimization.
>
> We view methods like DINO, iBOT, I-JEPA, and others as promising directions for future work. BEAST's MAE-based approach provides a strong baseline that establishes the value of domain-specific pretraining for behavioral videos, against which these more sophisticated methods can be compared and potentially improved upon.

---

### Official Review · Reviewer_gTWQ · 2025-10-31

**Soundness:** 3
**Presentation:** 3
**Contribution:** 3
**Rating:** 6
**Confidence:** 3

**Summary:**

The paper introduces BEAST (BEhavioral Analysis via Self-supervised pretraining of Transformers), a domain-tailored framework that pretrains experiment-specific ViTs on unlabeled animal-behavior videos by combining masked autoencoding with a temporal contrastive objective. A key ingredient is a frame selection/sampling strategy designed for static backgrounds with movement-driven variation. The pretrained backbone is evaluated on three tasks across multiple datasets/species. BEAST delivers competitive or superior results to strong baselines while reducing reliance on large labeled sets or keypoint-first pipelines.

**Strengths:**

1. Cleverly tailors self-supervised pretraining to behavioral videos (static background, movement-driven variation) via selection/sampling strategies.
3. Evaluates frozen-backbone features on segmentation and zero-/few-shot neural encoding, with systematic ablations.
3. Clear writing.
4. On action segmentation, BEAST bypasses pose-estimation training while matching or exceeding keypoint-based systems, reducing months-long annotation pipelines.

**Weaknesses:**

1. Cross-domain generalization. The approach emphasizes single-paradigm pretraining; comprehensive tests across cameras/environments/species would strengthen claims of generality.
2. Efficiency. The paper positions BEAST as more efficient than native video models but does not report controlled FLOPs/memory/runtime.
3. Interpretability. Beyond showing CLS superiority for pretraining, adding feature visualizations or sensitivity analyses linking learned features to anatomical/behavioral elements would aid scientific adoption.

**Questions:**

see weakness

---

> ### Author Response · Authors · 2025-11-21
>
> We thank the reviewer for recognizing the strengths of our work, and try to clarify any remaining concerns with our comments below.
>
> **W1: Cross-domain generalization.**
> We have demonstrated BEAST's state-of-the-art performance across diverse experimental contexts: head-fixed mice (IBL, IBL-whisker, Facemap, mirror-mouse), freely moving and socially interacting mice (CRIM13, CalMS21), and multi-view setups across two species (mirror-mouse, mirror-fish). These results provide strong evidence for BEAST's effectiveness across a range of experimental paradigms; however, we acknowledge that demonstrations across a broader range of environments and species would strengthen claims of generality. We will clearly note this limitation in the Discussion of the revised manuscript. We are excited to apply BEAST to additional species, multi-animal experiments, and multi-view setups, and believe our current work provides a strong foundation for this future research.
>
> **W2: Efficiency.**
> We thank the reviewer for this suggestion, as proper benchmarking significantly strengthens our efficiency claims. We recorded three metrics for each model: runtime (ms per batch), Giga Floating Point Operations per Second (GFLOPS), and memory required for a forward pass with batch size one. All experiments were run using the "fvcore" library on a single A100 GPU. Since runtime varies across batches while GFLOPS and memory are deterministic, we report mean and standard deviation of runtime across 32 batches.
>
> We benchmark two modes: "pretrain", which uses patch masking (0.75 for image models, 0.9 for VideoMAE, following optimal settings from the respective papers); and "finetune", which omits patch masking as relevant for our downstream tasks.
>
> | Model     | RUNTIME (ms/batch) |            | GFLOPS   |          | MEMORY (MB) |          |
> |-----------|-------------------|------------|----------|----------|-------------|----------|
> |           | Finetune          | Pretrain   | Finetune | Pretrain | Finetune    | Pretrain |
> | ViT-M     | 5.43±0.70         | 7.6±2.43   | 17.59    | 9.80     | 369.0       | 332.72   |
> | BEAST     | 6.21±0.94         | 7.9±2.26   | 17.59    | 9.80     | 409.1       | 333.40   |
> | VideoMAE  | 71.51±3.50        | 20.16±2.47 | 199.49   | 35.24    | 2955.1      | 831.61   |
>
> BEAST and ViT-M show comparable performance, with BEAST having a slightly longer runtime and larger memory footprint due to the nonlinear projector used in the contrastive loss (not substantial enough to affect GFLOPS). VideoMAE, however, requires substantially more resources due to processing 16 consecutive frames per batch element: during pretraining, it requires 2.5x runtime, 3.5x GFLOPS, and >2x memory compared to BEAST. These differences are even more pronounced during finetuning when patches are not masked: VideoMAE requires 11x runtime, 11x GFLOPS, and 7x memory.
>
> While VideoMAE and other video-based models are powerful, we have shown they require substantially increased resources for training and inference. This motivated our exploration of more efficient approaches like BEAST that will be more readily adoptable by individual labs in their data processing pipelines.
>
> **W3: Interpretability.**
> We agree with the reviewer that improved interpretability could be advantageous for addressing certain scientific questions. To explore this, we implemented a visualization strategy similar to the DINO series of papers. For a single image, we (1) extract patch embeddings, which are 768-dimensional vectors for each non-overlapping 16×16 (ViT-MAE/BEAST) or 14x14 (DINOv2) pixel patch; (2) perform PCA on these embeddings; and (3) visualize the top 3 principal components using RGB color channels. We apply this analysis to DINOv2, ViT-MAE (pretrained on ImageNet), and BEAST across multiple images per dataset. Results are shown in Appendix H.1.
>
> All methods capture spatial structure, but in qualitatively different ways. DINOv2 features tend to capture larger, more global image regions, whereas ViT-MAE and BEAST capture finer-grained local features. We also observe that DINOv2 features can include more background noise in some datasets (e.g., CRIM13, mirror-fish). Notably, BEAST's domain-specific pretraining does not fundamentally alter the ViT-MAE feature structure but appears to reduce noise. Further exploration of these visualizations will help elucidate the effects of our pretraining methods on learned representations.

---

> > ### Author Response · Authors · 2025-11-26
> >
> > We again thank the reviewer for their helpful feedback. We have expanded the Discussion to address the reviewer’s comment on cross-domain generalization (highlighted in blue in the updated pdf).

---

### Official Review · Reviewer_6xX2 · 2025-10-31

**Soundness:** 4
**Presentation:** 4
**Contribution:** 2
**Rating:** 4
**Confidence:** 4

**Summary:**

The authors propose BEAST, a transformer model for behavioral analysis with neuroscientific applications. The model uses a combination of a masked and contrastive objective on raw video frames, and can be used for three downstream tasks (neural encoding, action segmentation, pose estimation). The authors claim benefits in neural encoding over baseline methods for video/behavior encoding, improved zero shot neural encoding performance over a ViT baseline, improvements in keypoint estimation over a ViT baseline, and in action segmentation vs. the SimBA model. The supplementary material contains a number of additional experiments, variations and ablations of the proposed approach.

**Strengths:**

- Ablations and comparisions are very extensive and clearly outlined
- Paper is very well written. Dense, but very informative and to the point.
- The figures have high quality and are far above the average of ICLR papers
- The model has convincing performance across the different tasks.

**Weaknesses:**

While the method presented seems powerful for a variety of tasks, the evaluation as-is is currently too weak; it seems like BEAST is largely building on existing video-pre-training schemes. This in itself might be fine, but from e.g. Table 1 and 4 it seems that even frozen backbone models are suitable for solving the neural encoding tasks.

The authors need to better delineate what their methodological contribution is, and what it adds over a strong baseline model. I would e.g. consider doing a bit of finetuning with DINOv2, or VideoMAE, or TCN, on the authors' dataset, and check if this might surpass the performance of BEAST on one of multiple of the tasks; the tables suggest that this is likely. The paper might still be a nice contributions, but should be rephrased slightly; right now it appears like the exact combination of training objective is causing a performance gains, which I am critical about.

Some weaknesses I see are:

1. Figure 2: Choice of baselines is questionable. Motion energy and and PCA are good as "naive" baselines, but the CEBRA result feels obvious given that the backbone for BEAST is a ViT model, and for CEBRA the authors used a small convnet. The better comparison would be to use the same backbone, and only vary the training objective. The authors should also pull the baseline model (VIT-M) in Table 1 into this figure/barplot for better contextualization what their particular training objective contributes to this task.
2. Figure 3: Same concerns regarding baselines. BEAST is compared against a ResNet-50 model. What about other pose estimation algorithms that are widely used, e.g. DeepLabCut, which is also referenced in the text? These would be relevant comparisons for practitioners.
3. More baselines for action segmentation could be discussed and benchmarked; right now the result seems anecdotal.
4. While the results in the figures are nice, it seems like the authors created these benchmark situations and evaluated; it would be good to discuss how the authors validated their baselines, and if there are any previously published benchmarks BEAST can be applied on (and please excuse if there is a supplementary result that I overlooked already doing this)
5. The authors cite "temporal contrastive learning (Chen et al., 2020)", which is not the right reference for temporal contrastive learning. TCL and PCL might be better algorithms to cite, or even CPC (2018).

**Questions:**

1. I might have missed this in the numerous appendix results, but is there an ablation on the role of the two training objectives (masking vs. contrastive learning)? Are both required for getting high performance?
2. The method seems incremental in terms of the pre-training objective, which is also transparently communicated in the appendix. Could the authors elaborate again what the most conceptually close prior algorithm to BEAST is, and clearly delineate what the changes are they added, and what the effect of this change is? This could also be made a bit clearer in the main paper.
3. Judging from the std/confidence intervals in most tables, the difference between ViT variants and BEAST is slim. Could the authors run stats on the different tables and communicate transparently which advances are actually signficant?
4. Is the keypoint benchmark, especially for the baseline model, in Figure 3 grounded in the literature? Same question for Figure 4.
5. For neural encoding, Table 4 suggests that frozen backbones perform almost on par with BEAST. What happens if e.g. VideoMAE would be finetuned on the same dataset? I find it likely that this might already surpass BEAST performance. Was such an experiment (using an existing state of the art pre-training paradigm on the same data) done?
6. There is no mentioning that the code (training and/or inference) and/or the model weights are planned to be open sourced. Is this planned?

Happy to discuss further.

---

> ### Author Response · Authors · 2025-11-21
>
> We thank the reviewer for recognizing the care we took designing experiments, writing the manuscript, and creating figures.
>
> **W1: Baselines for neural encoding/CEBRA.**
> We chose CEBRA for a baseline because it provided a well-known contrastive-only approach to compare with our combined MAE+contrastive BEAST approach. However, we agree with the reviewer that the small CNN backbone of CEBRA does not provide the most rigorous contrastive-only baseline. To address this we have conducted two additional experiments: (1) using pretrained ViT-MAE representations as input to CEBRA (to exploit the representational power of the ViT but utilize the contrastive objective defined by CEBRA), and (2) ablating BEAST's training objective to use only the contrastive loss.
>
> For the first experiment, we extracted CLS tokens from the pretrained ViT-MAE network and used these as input to CEBRA's default TCN backbone to produce new representations (ViT-CEBRA). This approach controls for the powerful feature extraction of the ViT backbone while isolating differences in how these representations are leveraged. We then trained session-specific neural encoding models using the same procedure as for other methods (including BEAST). We find that using pretrained visual tokens does not improve standard CEBRA performance on the IBL neural encoding task. For IBL-whisker, ViT-CEBRA shows modest improvement over baseline CEBRA but remains substantially below BEAST's performance.
>
> IBL neural encoding (R^2)
> RRR = Reduced Rank Regression
> TCN = Temporal Convolution Network
> |                |            IBL            |        IBL-Whisker        |
> |                |     RRR     |     TCN     |     RRR     |     TCN     |
> |----------------|-------------|-------------|-------------|-------------|
> |CEBRA           | 0.226±0.030 | 0.283±0.050 | 0.177±0.049 | 0.218±0.057 |
> |ViT-CEBRA       | 0.189±0.060 | 0.258±0.062 | 0.209±0.022 | 0.248±0.054 |
> |BEAST           | 0.277±0.076 | 0.337±0.103 | 0.138±0.029 | 0.317±0.083 |
>
> For the second experiment we wanted to more precisely isolate the effects of the MAE versus temporal contrastive loss in BEAST. See the response to Q1 below for more information on this experiment.
>
> We agree with the reviewer that including ViT-M into Fig. 2 would provide important context, as well as the contrastive-only model we have now trained. However, rather than include two new models in Fig. 2 (which will be too much information for an already dense figure) we will add the contrastive-only model results to Table 1.
>
> **W3: Baselines for action segmentation.**
> We have attempted to provide extensive baselines for action segmentation by systematically varying the data representation while using a strong classification architecture from the literature.
>
> For the classification model, we use a temporal convolutional network (TCN), which was previously shown to substantially outperform linear classifiers and tree-based methods (XGBoost, Random Forests) on animal action segmentation tasks (Blau et al., 2025). We note that tree-based methods are used in standard animal action segmentation packages like JAABA (Kabra et al., 2013), MARS (Segalin et al., 2021) and SimBA (Goodwin et al., 2024), making the TCN a stronger baseline than what is typically used in the field.
>
> For data representation, we evaluate a range of keypoint-based and frame-based representations (Table 16). We agree that fine-tuning VideoMAE and DINOv2 would provide valuable additional baselines. We are currently training these models and will provide an update as soon as results are available.
> We welcome any additional baseline suggestions from the reviewer and are happy to explore other reasonable options within the rebuttal time window.
>
> **W4: Baseline validations.**
> The field of animal behavior analysis lacks the breadth and depth of benchmarks available in more established fields like computer vision. To compensate for this, we have exclusively used publicly available datasets (see public links in Appendix A) and have taken care to describe our baselines for each downstream task with extensive hyperparameter tuning (Appendices C, D, E).
>
> For each task, we leverage established methods from the literature: for neural encoding, we use the well-known CEBRA package (see previous comment regarding additional CEBRA experiments); for pose estimation, we use the Lightning Pose package and are currently implementing DeepLabCut as an additional baseline. For action segmentation, we evaluate on Calms21, which is an established benchmark dataset. Our ensembled BEAST model achieves an F1 of 0.84, placing in the top 15 of the challenge leaderboard (https://www.aicrowd.com/challenges/multi-agent-behavior-representation-modeling-measurement-and-applications/problems/mabe-task-1-classical-classification/leaderboards; top score: 0.89). We will contextualize our Calms21 results relative to the challenge leaderboard in the updated manuscript.

---

> > ### Author Response · Authors · 2025-11-21
> >
> > **W5: temporal contrastive learning references.**
> > We thank the reviewer for bringing these references to our attention. We will add these references to the “Self-supervised learning for images and videos” section of the Related Work.
> >
> > **Q1: Ablation on the role of the two training objectives.**
> > This is an excellent suggestion, and one that will greatly improve the completeness of our ablation studies. To address this, we pretrained a model on the IBL data using only the contrastive loss (but with patch masking, to make the comparison to the other models that use masked autoencoding more straightforward). We then evaluated the pretrained model on the neural encoding task. We find the mask-only model outperforms the contrastive-only model across both linear (RRR) and nonlinear (TCN) probes, for both IBL and IBL-whisker datasets, while the combined loss for BEAST remains the best performer (except for the linear probe in the IBL-whisker dataset).
> >
> > |                  |       IBL        |       IBL      | IBL-whisker | IBL-whisker |
> > |---------------|-----------------|---------------|---------------|-----------|
> > | Method        |     RRR     |     TCN     |     RRR     |     TCN     |
> > | Contrast only | 0.142±0.073 | 0.321±0.099 | 0.127±0.033 | 0.286±0.055 |
> > | Mask only     | 0.182±0.071 | 0.334±0.098 | 0.156±0.032 | 0.316±0.073 |
> > | Combined      | 0.277±0.076 | 0.337±0.103 | 0.138±0.029 | 0.317±0.083 |
> >
> > We are now working to evaluate the contrastive-only model on the pose estimation task.
> >
> > **Q2: methodological novelty of BEAST.**
> > The most conceptually close prior algorithm to BEAST is the ViC-MAE (Hernandez et al., 2024 ECCV). We highlight the most significant change in L126-133:
> > ```
> > The most significant modification is how frames are sampled for the contrastive loss. ViC-MAE allows any two frames from the same video to be a positive pair, and frames from different videos are negative pairs (Xu and Wang, 2021). While this may be appropriate for benchmark datasets with short clips, animal behavior experiments generate long-duration recordings where behaviors repeat across time. We instead define positive frames within a narrow temporal window around the anchor (±1 frame), while allowing negative frames to be either distant and dissimilar frames in the same video, or from different videos. Crucially, this strategy outperforms that of ViC-MAE (Table 6).
> > ```
> > A more minor modification is the input we use to the nonlinear projector used in the temporal contrastive loss: ViC-MAE uses a pooled attention layer to transform patch embeddings, while BEAST uses the CLS token (described in Appendix B.1). Using the CLS token is both architecturally and conceptually simpler and leads to performance improvements, especially for the linear probe of the IBL data (Table 8).
> >
> > We note that one of our core contributions lies in how established self-supervised objectives (masked autoencoding and temporal contrastive learning) are adapted and applied in a domain-specific manner to address longstanding challenges in behavioral neuroscience. BEAST is, to our knowledge, the first framework to perform self-supervised pretraining on animal behavior videos across multiple subjects, and then reuse the resulting backbone for diverse downstream tasks such as neural encoding, pose estimation, and action segmentation. While the constituent losses are not novel, our findings show that adapting these techniques to the behavioral neuroscience domain, in combination with a tailored contrastive sampling strategy and careful experimental design, yields meaningful performance improvements and substantial practical benefits. We hope the reviewer sees value in this form of contribution, particularly in light of the clear gap in accessible, general-purpose models for scientific video analysis.
> >
> > **Q4: Pose estimation and action segmentation benchmarks.**
> > For pose estimation, we use four datasets made publicly available by the authors of the Lightning Pose paper (Biderman et al., 2024; see Appendix A for data links). Our ResNet-50 baseline replicates the Lightning Pose training setup and hyperparameters (batch size, learning rate, epochs, pretrained backbone, backbone unfreezing epoch). We are also currently implementing DeepLabCut as an additional baseline to better connect our results to the animal pose estimation literature.
> >
> > For action segmentation, we refer the reviewer to our responses above (“W3: Baselines for action segmentation” and “W4: Baseline validations”).
> >
> > **Q6: Open sourcing code and model weights.**
> > We plan to open source all pretraining code (frame selection, model training and inference) in a Github repository upon acceptance of this manuscript. We will also deposit pretrained models for all datasets on Hugging Face.

---

> ### Author Response · Authors · 2025-11-21
>
> **Work in progress.**
> We are still working on responding to the following comments:
> * W2: Baselines for pose estimation/DeepLabCut
> * W3: Baselines for action segmentation/DINOv2
> * Q3: Neural encoding statistics
> * Q5: VideoMAE pretraining

---

> ### Author Response · Authors · 2025-11-26
>
> **W5: temporal contrastive learning references.**
> We have changed the reference in the Introduction to Hyvarinen and Morioka 2016 (thanks again for bringing this reference to our attention).
>
> We have also added the following lines to the Related Work section (highlighted in blue): “The contrastive method has also been extended to the temporal l (Hyvarinen and Morioka, 2016) and video domain (Qian et al., 2021; Recasens et al., 2021; Dave et al., 2022).”
>
> We also cite CPC in the Methods section under the “Temporal contrastive loss” paragraph when we refer to the InfoNCE loss. If the reviewer feels we have missed any other relevant references we are happy to add them.
>
> **W2: Baselines for pose estimation/DeepLabCut.**
> Two reviewers requested DeepLabCut baselines, and we copy the same response to both reviewers for ease of reading.
>
> We thank the reviewers for this suggestion, as appropriate and widely used baselines are essential for demonstrating the efficacy of our approach. We trained DeepLabCut models on our pose estimation datasets using the default ResNet-50 backbone. To isolate differences in the underlying algorithms, we matched training frames, batch size, learning rate schedule, and number of epochs with our Lightning Pose models. For all other hyperparameters, we used the respective package defaults.
>
> Lightning Pose outperforms DeepLabCut on the IBL, Mirror-mouse, and Mirror-fish datasets, while DeepLabCut outperforms Lightning Pose on CRIM13. Notably, our pretrained BEAST backbones significantly outperform both DeepLabCut and Lightning Pose across all four datasets. We provide a summary of results below (mean plus/minus standard error of the mean across all test keypoints/frames), with full pixel error versus ensemble standard deviation plots in Appendix H.5 (Fig. 14) and reference to this baseline in the main text.
>
> | Model           | IBL         | Mirror-mouse | CRIM13       | Mirror-fish  |
> |-----------------|-------------|--------------|--------------|--------------|
> | resnet50 (DLC)  | 11.86 ± 0.4 | 9.61 ± 0.42  | 18.13 ± 0.23 | 16.14 ± 0.46 |
> | resnet50 (LP)   | 8.4 ± 0.25  | 6.25 ± 0.18  | 21.72 ± 0.2  | 10.39 ± 0.23 |
> | BEAST           | **5.77 ± 0.15** | **4.72 ± 0.13**  | **16.02 ± 0.12** | **7.34 ± 0.12**  |

---

> > ### Author Response · Authors · 2025-11-26
> >
> > **W3: Baselines for action segmentation/DINOv2.**
> > Two reviewers were interested in DINOv2 results for action segmentation, so we copy the same text to each for ease of reading.
> >
> > We thank the reviewers for pointing out this gap in our baselines. We have now fine-tuned a DINOv2 model on action segmentation using patch embeddings, following the same procedure as for our other pretrained ViT backbones. We find that DINOv2 performs worse (F1 scores, higher is better) than both the ImageNet-pretrained ViT-MAE and BEAST:
> >
> > | Model           | IBL         | CalMS21      |
> > |-----------------|-------------|--------------|
> > | ViT-M (IN)      | 0.84 ± 0.00 | 0.74 ± 0.00  |
> > | DINOv2          | 0.77 ± 0.00 | 0.68 ± 0.00  |
> > | BEAST   | **0.87 ± 0.01** | **0.81 ± 0.01**  |
> >
> > We used identical hyperparameters across all three models due to computational constraints. DINOv2's performance may improve with additional hyperparameter tuning, which we will continue to explore within the remaining rebuttal period.
> > With this result, we have now incorporated the DINOv2 baseline across all downstream tasks. This comprehensive comparison strengthens our claims about the benefits of domain-specific pretraining for behavioral neuroscience video data.
> >
> > Summary of BEAST vs. DINOv2 across tasks:
> > - **Neural encoding**: BEAST slightly outperforms DINOv2 (Table 4)
> > - **Pose estimation**: BEAST significantly outperforms DINOv2 on 3 of 4 datasets (Fig. 7)
> > - **Action segmentation**: BEAST significantly outperforms DINOv2 on both datasets (Table 16)
> >
> > Qualitative feature analysis (Fig. 10) reveals that DINOv2 features are visually distinct from those of MAE-based backbones (ImageNet-pretrained ViT-MAE and BEAST). Further work will be required to more thoroughly assess how different pretraining strategies affect performance on neuroscience-specific downstream tasks.
> >
> > **Q3: Neural encoding statistics.**
> > Two reviewers brought up concerns about significance in the neural encoding results. We appreciate this attention to detail and copy the same text for response to both reviewers for ease of reading.
> >
> > For the results in our tables we first computed the mean BPS per session (call this mBPS), then provide the mean and the standard deviation of the mBPS over five sessions for each dataset in the tables. This relatively small sample size of sessions leads to large standard deviations. To more carefully measure the differences between models on the full neural populations with the IBL dataset, we performed a two-sided Wilcoxon signed-rank test at the level of individual neurons (N=842 pairs) for the following models:
> > - DINOv2
> > - ViT-M (IN): a ViT pretrained on Image-Net with MAE loss
> > - ViT-C (IN+PT): ViT-M further pretrained on domain-specific data with the contrastive-only loss
> > - ViT-M (IN+PT): ViT-M further pretrained on domain-specific data with the MAE loss
> > - BEAST (IN+PT): ViT-M with additional domain-specific pretraining using MAE and contrastive losses
> >
> > The detailed results are in Appendix H.4: Figs. 12 and 13, and Table 17. We find that BEAST (IN+PT) generally significantly outperforms all other models, though it is not significantly different from ViT-M (IN+PT) on either dataset when using the TCN. Table 17 contains the recomputed means and standard errors over all neurons rather than session averages (i.e. what is shown in Table 14). We will continue to update Table 17 with more baselines throughout the rebuttal period.
> >
> > **Q1: ablation of the two training objectives using pose estimation for evaluation**
> > We previously showed ablation experiments on BEAST's training objective by training a contrastive-only model and evaluating it on the neural encoding task (see table above). We have now extended this analysis to the pose estimation task on the IBL dataset and find that the contrastive-only backbone performs considerably worse than the MAE-only backbone (Appendix H.6, Fig. 15). This result is consistent with the different learning objectives: the temporal contrastive loss emphasizes high-level temporal structure, whereas the MAE loss emphasizes low-level, pixel-level features. Consequently, MAE-pretrained representations are better suited for pixel-level prediction tasks like pose estimation, while the contrastive loss provides complementary benefits for tasks requiring temporal information (as evidenced by the improved neural encoding performance when both losses are combined).
> >
> > **Work in progress.**
> > We are still working on responding to the following comments:
> > - Q5: VideoMAE pretraining

---

> > > ### Author Response · Authors · 2025-12-03
> > >
> > > **Q5: VideoMAE pretraining.**
> > >
> > > We have now trained a VideoMAE model on the IBL dataset and evaluated it on the neural encoding task (Table 17). While further pretraining on IBL shows a very slight improvement in mean BPS over the frozen Kinetics-400-pretrained VideoMAE, two-sided Wilcoxon signed-rank tests reveal no significant differences among all pairs of the following three models (BPS $\pm$ SEM over N=842 test neurons): VideoMAE pretrained on Kinetics-400 (0.330 $\pm$ 0.013), VideoMAE further pretrained on IBL (0.334 $\pm$ 0.013), and BEAST (0.335 $\pm$ 0.013).
> > >
> > > These results demonstrate that while video-based self-supervised learning methods like VideoMAE are viable for behavioral neuroscience data, they are matched in performance by BEAST, which is substantially more computationally efficient (Table 9). BEAST achieves comparable results by incorporating temporal information through a contrastive loss rather than processing multiple frames through an MAE-style objective. Nevertheless, we view video-based pretraining as an important direction for future work.

---

### Author Response · Authors · 2025-11-21

We sincerely thank the reviewers for their thoughtful evaluations and constructive feedback. We believe that addressing these points has already strengthened our submission significantly.

Below we address the comments of each individual reviewer. We have organized our responses into three categories:

1. **Minor revisions**: For straightforward concerns (e.g., adding missing references, clarifying writing), we acknowledge the feedback and indicate the changes we will make in the final version before the rebuttal period is over.

2. **Substantive responses with new results**: For more in-depth comments requiring additional experiments, we have added a new section to the appendix of our manuscript (Appendix H). We will post initial conclusions in our text responses here and direct you to specific sections of the updated PDF for detailed results and visualizations. We will continue to update this manuscript as we complete additional analyses.

3. **Work in progress**: For comments we are still actively addressing, we list these at the end of each reviewer response and will post follow-up comments as new results become available.

We plan to update manuscript revisions iteratively over the coming weeks and will notify you when significant new results are added.

We hope this structure will assist in your evaluation and look forward to continuing discussions with each of you.

---

### Author Response · Authors · 2025-12-03
**Summary of updates**

We sincerely thank the reviewers again for their time and constructive feedback. We regret that the discussion period ended prematurely, and have worked hard to thoroughly address all remaining comments and questions.

**Positive feedback shared across reviewers:**
* Our vision transformer pretraining strategy, BEAST, demonstrates convincing performance across a wide range of downstream tasks under realistic settings.
* We conduct extensive ablations and compare to numerous baselines.
* The paper is well-written with high-quality figures.

Responding to reviewers' feedback has substantially strengthened our submission through additional baselines, ablations, and references. Below we summarize the main updates, which are highlighted in blue text in the updated manuscript.

**New Baselines**:
* **DINOv2** is now implemented across all three downstream tasks: neural encoding (Table 17), pose estimation (Fig. 7), and action segmentation (Table 16). BEAST consistently outperforms this powerful general-purpose backbone (except for one pose estimation dataset).
* **DeepLabCut** is now implemented for pose estimation. BEAST significantly outperforms DeepLabCut across all datasets (Fig. 14).
* **VideoMAE** (pretrained on Kinetics-400) was further pretrained on the IBL dataset. Its performance is not significantly different from BEAST (Table 17), but requires significantly longer training and inference times (Table 9).

**Training objective ablation**: The original submission compared BEAST (MAE + contrastive) with MAE-only models. We now include contrastive-only ablation results across all three tasks on the IBL dataset: neural encoding (Table 1), pose estimation (Fig. 15), and action segmentation (Table 16). We consistently find that combining both losses outperforms either loss alone. Due to the limited rebuttal timeframe, we performed this ablation on one dataset that spans all three downstream tasks; we are already working to complete this ablation across all remaining datasets for the camera-ready version.

**Statistical Testing**: We performed two-sided Wilcoxon signed rank tests on neural encoding results (Figs. 12, 13; Table 17). BEAST significantly outperforms DINOv2, ImageNet-pretrained ViT (ViT-MAE), and single-loss BEAST variants (except for ViT-M with nonlinear probes on IBL/IBL-whisker datasets, where differences are not significant). We will incorporate these significance tests throughout the manuscript for the camera-ready version.

**Additional Improvements**:
* **Efficiency benchmarking**: Compared training and inference costs across ViT-MAE, BEAST, and VideoMAE (Table 9), demonstrating that VideoMAE requires considerably more time, FLOPS, and memory than BEAST.
* **Feature visualization**: Provided qualitative analysis of patch embeddings from DINOv2, ViT-MAE, and BEAST (Fig. 10), illustrating similarities and differences that can guide understanding of how pretraining objectives affect downstream task performance.
* **References**: Added missing citations for temporal contrastive loss, spatiotemporal contrastive loss on videos, and pose estimation in Related Work. Included a new Related Work section on self-supervised learning for animal behavior analysis (Appendix H.2).
* **Discussion**: Updated to address reviewer questions about cross-domain generalization and applying BEAST to naturalistic environments, and expanded on directions for future work.

The new figures and analyses are currently organized in "Appendix H: Reviewer Responses" to maintain consistent numbering throughout the review process. For the camera-ready version, we will integrate these figures and accompanying text into the appropriate sections of the main manuscript and standard appendix to ensure optimal organization and readability.

---

### Meta-Review · Area_Chair_gP9Y · 2025-12-18

**Summary:**

This paper introduces BEAST (BEhavioral Analysis via Self-supervised pretraining of Transformers), a framework that combines masked autoencoding (MAE) with temporal contrastive learning to pretrain vision transformers on animal behavior videos. The authors evaluate BEAST on three downstream tasks—neural encoding, pose estimation, and action segmentation—across multiple datasets and species. The reviewers collectively acknowledged the paper's strengths in experimental design, comprehensive evaluation, and clear presentation, while raising concerns about methodological novelty, baseline comparisons, and statistical significance. The authors provided an extensive rebuttal with substantial new experiments that significantly strengthened the submission.

**Reviewer Concerns:**

**Addressed by Rebuttal:**

* Baseline comparisons: All reviewers requested additional baselines. The authors added:
  * DINOv2 across all three downstream tasks, showing BEAST consistently outperforms this strong general-purpose backbone (except one pose estimation dataset)
  * DeepLabCut for pose estimation, where BEAST significantly outperforms across all datasets
  * VideoMAE pretrained on IBL dataset, showing comparable performance to BEAST but with substantially higher computational costs

* Statistical significance: Reviewers 6xX2 and 2yb7 noted that improvements appeared borderline significant given the error bars. The authors conducted two-sided Wilcoxon signed-rank tests at the neuron level (N=842 pairs), demonstrating BEAST significantly outperforms DINOv2, ImageNet-pretrained ViT, and single-loss variants in most configurations.
* Training objective ablation: Reviewer 6xX2 requested ablation of the contrastive-only loss. The authors trained and evaluated this model, showing the combined loss generally outperforms either loss alone.
* Efficiency benchmarking: Reviewer gTWQ requested controlled comparisons of computational costs. The authors provided detailed FLOPS, memory, and runtime measurements, demonstrating VideoMAE requires 11x more resources during finetuning.
* Missing references: Reviewers ys7M and 2yb7 identified missing citations. The authors added extensive references for temporal contrastive learning and created a new Related Work section on SSL for animal behavior analysis.

**Remaining Concerns:**
* Methodological novelty: Multiple reviewers (6xX2, ys7M, 2yb7) noted that BEAST's core components (MAE + contrastive learning) build directly on ViC-MAE. The authors acknowledge their primary contribution is domain-specific adaptation rather than architectural innovation. While they argue this adaptation provides significant practical value to behavioral neuroscience, the incremental nature of the technical contribution could still be perceived as a limitation.
* Performance margins: Despite statistical significance testing, some performance improvements remain modest. The authors provide appropriate caveats, but this limits claims about the necessity of the combined objective.

**Reviewer Scores:**

* Reviewer 6xX2: 4 (marginally below acceptance) - Did not update score despite extensive new experiments. The AC attributes this to the frozen discussion due to the OpenReview bug. Given that the reviewer indicated openness to discuss at the end their review, the AC believes that the substantial replies by the authors would have led the reviewer to increase their score.
* Reviewer gTWQ: 6 (marginally above acceptance)
* Reviewer ys7M: 6 (marginally above acceptance) - Explicitly stated satisfaction with responses and maintained rating
* Reviewer 2yb7: 6 (marginally above acceptance) - Engaged substantively but did not provide final updated score, likely also because of the OpenReview bug. Thus the AC believes that the reviewer would at least has kept their score.

---

### Decision · Program_Chairs · 2026-01-26

Accept (Poster)